# Filamentous recombinant human Tau activates primary astrocytes via an integrin receptor complex

Peng Wang[1] & Yihong Ye [1 ✉]

Microtubule-associated protein Tau can form protein aggregates transmissible within the brain, correlating with the progression of tauopathies in humans. The transmission of aggregates requires neuron-released Tau to interact with surface receptors on target cells. However, the underlying molecular mechanisms in astrocytes and downstream effects are unclear. Here, using a spatially resolved proteomic mapping strategy, we show that integrin αV/β1 receptor binds recombinant human Tau, mediating the entry of Tau fibrils in astrocytes. The binding of distinct Tau species to the astrocytic αV/β1 receptor differentially activate integrin signaling. Furthermore, Tau-mediated activation of integrin signaling results in NFκB activation, causing upregulation of pro-inflammatory cytokines and chemokines, induction of a sub-group of neurotoxic astrocytic markers, and release of neurotoxic factors. Our findings suggest that filamentous recombinant human Tau-mediated activation of integrin signaling induces astrocyte conversion towards a neurotoxic state, providing a mechanistic insight into tauopathies.

[1] Laboratory of Molecular Biology, National Institute of Diabetes and Digestive and Kidney Diseases, National Institutes of Health, Bethesda, MD 20892, USA.
✉email: yihongy@mail.nih.gov

Clinically documented protein aggregates such as extracellular amyloid plaques and intracellular neurofibrillary tangles (NFTs) are pathological hallmarks of neurodegenerative diseases such as Alzheimer's disease (AD). NFTs are formed primarily by hyper-phosphorylated Tau, a soluble microtubule-binding protein abundantly expressed in neurons of the central nervous system (CNS)[1,2]. The primary function of Tau is to stabilize microtubules, but its intrinsic misfolding propensity prompts it to form toxic filamentous aggregates, particularly in neurons during aging. In addition to AD, Tau-containing NFTs have been linked to brain dysfunctions in other neurodegenerative diseases such as Pick's disease and frontotemporal dementia, which are collectively termed Tauopathies[1].

An unusual characteristic of tauopathies is the prion-like propagation of Tau-containing aggregates, which seems to correlate with the decline of cognition during disease progression[1–6]. This protein propagation process may involve the release of Tau in either monomeric or aggregated forms, followed by its uptake by recipient cells (neurons or glial cells)[7]. Tau fibrils formed along this path can serve as "seeds" once inside a target cell, converting endogenous Tau monomers into an aggregation-prone toxic conformer. Accumulating evidence suggests that both monomeric Tau and Tau aggregates can be released from neurons independent of cell death, and this process is enhanced by neuronal activities[8–10]. The mechanisms underlying Tau release are controversial. Specifically, some studies showed that Tau is predominantly released in a free soluble form[11–14], but a study suggested membrane-associated vesicles as a major extracellular Tau carrier[8]. For Tau internalization, numerous studies showed that neurons, microglia, and astrocytes all could readily internalize different Tau species[15–18]. In certain immortalized cells, endocytosis of Tau preformed fibrils (PFFs) is initiated when Tau binds to heparan sulfate proteoglycans (HSPGs) on the cell surface[19–21], which cooperate with a membrane receptor to mediate Tau internalization[22], but other unidentified mechanisms also contribute to Tau uptake, particularly in primary non-neuronal cells[22,23].

In addition to damaging endogenous Tau folding when Tau PFFs enter neurons, Tau fibrils can also influence non-neuronal cells from the cell exterior. This model is particularly attractive in the complex brain microenvironment in which the fitness of neurons is tightly modulated by glial cells such as astrocytes[24,25]. It was shown that the activation of astrocytes by lipopolysaccharide (LPS) can alter their phagocytic activity, which might mitigate Tauopathies as activated astrocytes help to clear protein aggregates[26,27]. However, activated astrocytes might also exacerbate Tauopathies because they can contribute to neuroinflammation by releasing pro-inflammatory cytokines and chemokines[28,29]. Notably, a recent transcription profiling study revealed two distinct functional states of reactive astrocytes termed as "A1" and "A2", respectively[30]. A2 astrocytes appear to restore neuronal activities after injury, whereas A1 astrocytes not only fail to promote synapse formation, but also gain a neurotoxic activity by releasing some uncharacterized factors.

How astrocytes are converted to a neurotoxic state is a key question relevant to many neurodegenerative diseases. To date, accumulating evidence suggest that microglia activated during the classic inflammation response can serve as a mediator of astrocytic A1 conversion, because factors released by reactive microglia are sufficient to activate the astrocyte A1 state[30]. However, whether neurotoxic astrocytes can be induced independent of microglia under stress or disease conditions is unclear.

Given the close link between Tau misfolding and neurodegenerative diseases, we investigate whether harmful insults triggered by extracellular Tau aggregates can directly convert astrocytes into a neurotoxic state. We used a proximity-based ligation approach to identify the αV/β1 integrin complex as a receptor for Tau monomer and fibrils in primary astrocytes. The ligation of Tau fibrils to integrin not only initiates their internalization, but also activates integrin signaling, which in turn induces inflammation and converts astrocytes into an A1-like neurotoxic state. These findings establish a direct link between misfolding-associated proteotoxic stress and astrocyte activation.

## Results

**HSPG-independent uptake of Tau PFFs by primary astrocytes.** To study the mechanism of Tau fibril uptake in primary astrocytes, we purified Tau (2N4R) as a recombinant protein from *E. coli* (Supplementary Fig. 1a) and labeled it with a fluorescent dye (Alexa$_{594}$). We used the labeled protein to generate PFFs following a well-established protocol (Supplementary Fig. 1b)[31], and then treated mouse primary astrocytes with labeled monomeric Tau or PFFs. Incubating astrocytes with Tau for up to 6 h did not cause detectable cell death (Supplementary Fig. 1c) but led to efficient internalization of Tau PFFs. Unlike immortalized cell lines, Tau PFF uptake by primary astrocytes was largely unaffected by heparin (Supplementary Fig. 1d, e), a competitive inhibitor that blocks Tau PFF binding to HSPGs in some cell lines[19–21]. This result is in line with another study, which suggests that primary astrocytes do not use HSPGs for Tau PFF internalization[32]. Instead, an unidentified Tau receptor(s) is involved.

**Identification of integrin αV/β1 as a Tau interactor in primary astrocytes.** Because Tau monomers could also enter primary astrocytes, albeit with reduced efficiency compared to Tau PFFs[15,32], we presumed that Tau fibrils and monomers engaged a common receptor but with different affinities. Therefore, we employed a proximity labeling-based proteomic mapping strategy using purified APEX2-tagged Tau monomer as the bait (Fig. 1a). APEX2 is an engineered peroxidase that can covalently tag proximal lysine side chains with biotin molecules in the presence of biotin-phenoxyl radicals[33]. Because the relatively large APEX2 fusion might affect the interaction of Tau with the plasma membrane, we first used Sh-SY5Y cells to characterize the membrane interaction of Tau fusion proteins (Fig. 1b). When APEX2 was fused to the C-terminus of Tau, Tau-APEX2 bound efficiently to the cell surface in a dose dependent manner, but the addition of APEX2 to the N-terminus abolished the binding (Supplementary Fig. 2a, b). Thus, consistent with previous reports[34,35], Tau binding to the plasma membrane is mediated by a sequence near the N-terminus, which can be masked by a proximal APEX2 fusion.

To identify Tau receptor, we incubated Tau-APEX2 or APEX2 as a control with primary astrocytes at 4 °C for 30 min to allow membrane binding. This short incubation did not affect membrane permeability or induce cell death (Supplementary Fig. 2c). We then performed biotin labeling and the labeled plasma membrane proteins were enriched by streptavidin-affinity purification. Immunoblotting using biotin antibodies detected a major biotinylation product with Mr. of ~120 kDa (Fig. 1c). As expected, APEX2 alone was not able to biotinylate this 120 kDa protein (Fig. 1c). Mass spectrometry analysis of the excised band identified several membrane proteins enriched in Tau-APEX2-treated cells (Supplementary Fig. 2d). Interestingly, the top candidates were all cell adhesion molecules such as integrins and neural cell adhesion molecules (NCAM).

The identification of integrin as a potential Tau receptor was intriguing given that genetic variations in integrin genes had been suggested as a potential risk factor for neuroinflammation/neurodegeneration[36–38] and that α6/β1 integrin was shown to facilitate phagocytosis-mediated elimination of Aβ amyloid[39,40].

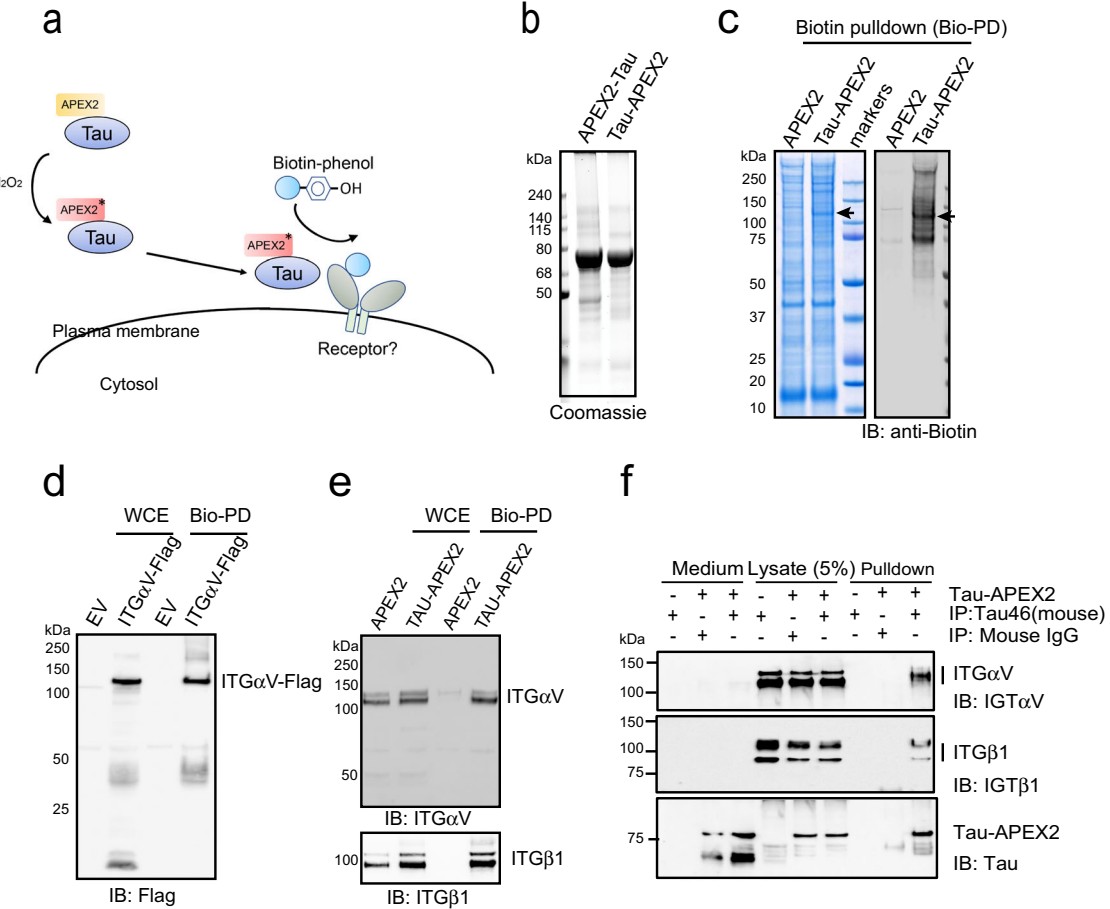

**Fig. 1 Identification of integrin αV/β1 as a Tau interactor in primary astrocytes. a** Schematic overview of the proximity-based proteomic mapping strategy. Tau-APEX2 (200 nM) is bound to the cell surface. Biotinylation was initiated by incubating cells with biotin-phenol (BP) in the presence of $H_2O_2$ for 4 min. Biotinylated proteins were then purified by affinity chromatography using streptavidin resin. **b** Coomassie blue staining of purified Tau proteins. **c** Coomassie blue staining (left) and immunoblotting (IB) (right) of biotinylated proteins from primary astrocytes incubated with APEX2 or Tau-APEX2. The arrow indicates a 120 kDa protein detected in Tau-APEX2-treated cells. **d**, **e** Validation of Tau-APEX2-mediated biotinylation of ITGαV/β1. **d** Tau-APEX2 was incubated with HEK293T cells transfected with either an empty vector (EV) or ITGαV-Flag. Whole cell extracts (WCE) were either directly analyzed by immunoblotting or first subjected to biotin pulldown (Bio-PD) before immunoblotting. **e** Whole cell extracts (WCE) from primary astrocytes treated with either APEX2 or Tau-APEX2 were analyzed directly by immunoblotting or first subjected to biotin pulldown before immunoblotting. **f** Co-immunoprecipitation of Tau-APEX2 with ITGαVβ1. Astrocytes were treated with Tau-APEX2 (200 nM) or PBS as a control for 30 min on ice. Cells were lysed in an NP40-containing lysis buffer. Cell lysates were subjected to immunoprecipitation with either a control IgG or a Tau specific antibody (Tau-46). Both cell lysates and precipitated proteins were analyzed by immunoblotting. Where indicated, a fraction of the remaining medium was also analyzed by immunoblotting to show input.

Since αV/β1 integrin (ITGαV/β1) was the most significantly modified proteins (based on peptide count), we focused this study on ITGαV/β1.

To validate the interaction of Tau-APEX2 with αV/β1 integrin, we repeated Tau-APEX2-mediated biotinylation using HEK293T cells over-expressing either Flag-tagged ITGαV or GFP-tagged ITGβ1. Following cell surface biotinylation and streptavidin pulldown, these proteins could be detected by immunoblotting using antibodies recognizing their corresponding tags (Fig. 1d and Supplementary Fig. 2e), suggesting that they could indeed be biotinylated by Tau-APEX2. Similar experiments performed in primary astrocytes further confirmed that endogenous ITGαV/β1 could be biotinylated by Tau-APEX2 but not APEX2 (Fig. 1e). Importantly, after Tau-APEX2 binding to the cell surface, immunoprecipitation of Tau-APEX2 from cell extracts using a Tau specific antibody also co-precipitated endogenous ITGαV/β1 (Fig. 1f). Likewise, when Tau-APEX2 was incubated with purified ITGαV/β1 ectodomains, immunoprecipitation of Tau also co-precipitated ITGαV/β1 (Supplementary Fig. 3a).

Thus, the biotinylation of ITGαV/β1 by Tau-APEX2 is likely a result of a direct interaction between these proteins.

**αV/β1 integrin mediates Tau PFF uptake in primary astrocytes.** We next tested whether Tau PFFs also interacted with endogenous ITGαV/β1 by co-immunoprecipitation. To this end, we incubated primary astrocytes in the presence or absence of Tau PFFs on ice before solubilizing cells by a NP40-containing buffer. Immunoprecipitation using Tau antibodies precipitated Tau PFFs together with a fraction of endogenous ITGαV/β1 (Fig. 2a, lane 9). By contrast, ITGαV/β1 was not precipitated when Tau PFF was omitted or when a control antibody was used (lanes 7, 8). We conclude that ITGαV/β1 also interacts with Tau PFFs and thus may serve as a receptor for both monomeric and filamentous Tau.

We next tested whether ITGαV/β1 mediates Tau PFF cell entry. We generated lentiviruses expressing small hairpin RNAs (shRNAs) that specifically knocked down *ItgαV* or *Itgβ1* (Fig. 2b, c). Knockdown of *ItgαV* and *Itgβ1* either individually or in

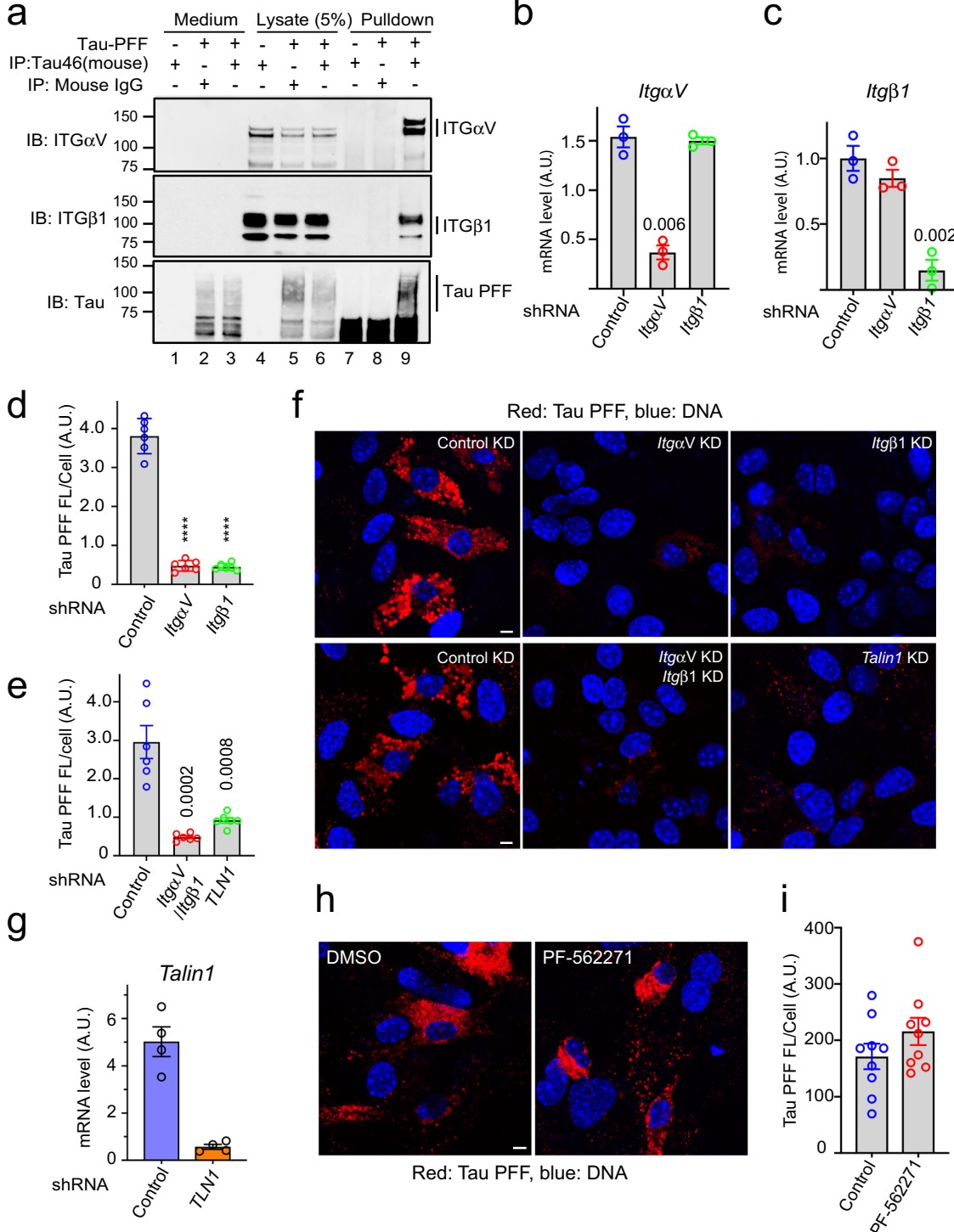

combination did not affect the viability of astrocytes, but significantly reduced Tau PFF internalization (Supplementary Fig. 3b and Fig. 2d–f). Furthermore, knockdown of *Talin1*, an adapter essential for integrin signaling and integrin-mediated endocytosis[41–43] also led to a significant reduction in Tau PFF uptake (Fig. 2e–g). The fact that knockdown of three different integrin pathway components all reduced Tau PFF uptake ruled out the possibility of an off-target effect. Intriguingly, treating cells with a potent inhibitor (PF-562271) of the focal adhesion kinase (FAK), an essential integrin regulatory kinase downstream of Talin1 did not affect Tau PFF endocytosis (Fig. 2h, i), nor did it

affect cell viability (Supplementary Fig. 3c). These results suggest that binding to ITGαV/β1 triggers Tau PFF cell entry, which requires Talin1 but not the downstream integrin signaling.

**Monomeric Tau and Tau PFFs differentially activate integrin signaling.** The ligation of ligands such as fibronectin to integrin activates integrin signaling, changing the expression of many downstream genes[44]. To test whether Tau also activated integrin signaling, we treated primary astrocytes with monomeric Tau, Tau PFFs, or as a control with phosphate buffer saline (PBS) for 6 h. qRT-PCR analyses of a collection of integrin target genes

**Fig. 2 Integrin αV/β1 and Talin1 are required for Tau PFF uptake in astrocytes. a** Co-immunoprecipitation of Tau PFF and ITGαV/β1. Astrocytes were treated with Tau PFFs or PBS as a control. Cell lysates were subjected to immunoprecipitation with either a control IgG or a Tau specific antibody (Tau-46). Both lysates and precipitated proteins were analyzed by immunoblotting. Where indicated, a fraction of the medium collected after the binding was also analyzed by immunoblotting. **b, c** Quantification (mean ± SEM) of mRNA expression by qRT-PCR performed 72 h postlentiviral infection confirmed the knockdown efficiency of ITGαV (**b**), β1 (**c**). A.U. arbitrary unit. $n = 3$ biologically independent experiments. The numbers above indicate $p$ values determined by two-tailed unpaired $t$-test. **d–g** Astrocytic uptake of Tau PFF is dependent on ITGαV/β1 and Talin1. **f** Representative images of lentivirus-treated astrocytes incubated with Alexa594 (Red)-labeled Tau-PFF (200 nM) for 2 h. Cells were stained with Hoechst to label the nuclei (blue). Scale bar, 5 μm. **d, e** Quantification of the relative Tau PFF fluorescence intensity/cell. Mean ± SEM, $n = 3$ biologically independent experiments. The numbers and asterisks above indicate $p$ values (****$p < 0.0001$) determined by two-tailed unpaired $t$-test. Dots represent the average fluorescence intensity/cell from images (>20 cells/image) acquired. **g** As in **b**, except that Talin1 mRNA expression was determined after Talin1 knockdown. Mean ± SEM, $n = 4$ biologically independent samples. **h, i** FAK is not required for Tau PFF uptake. **h** Representative images of DMSO or PF-562271-treated astrocytes (10 μM) incubated with labeled Tau-PFF (200 nM) after 2 h. Scale bar, 5 μm. **i** Quantification of Tau PFF uptake in **h**. Mean ± SEM, $n = 3$ biologically independent experiments. Dots represent average fluorescence intensity/cell from images acquired.

(*Mmp9*, *Icam1*, and *Mmp13* etc.)[45,46] showed that compared to PBS-treated cells, Tau monomer and PFFs differentially activated the expression of many integrin target genes with Tau PFF being ~2–3-fold more potent than monomeric Tau (Fig. 3a). As anticipated, PFF-induced integrin activation could be mitigated if astrocytes were first infected with lentiviruses that knocked down *ItgαV* and *Itgβ1* (Fig. 3b).

Because astrocytes purified by the conventional method also contain some microglia and oligodendrocytes, we used immunopanning-purified astrocytes to further validate Tau PFF-induced integrin signaling (Fig. 3c) (see the "Methods" section). Immunostaining using an antibody against the astrocyte-specific marker GFAP showed that purified astrocytes were free of other cell types. Immunopurified astrocytes internalized Tau PFFs similarly as conventionally purified astrocytes (Fig. 3c). These cells also activated integrin target genes after being stimulated by Tau PFFs (Fig. 3d), which was reproducibly diminished when Talin1 was depleted (Fig. 3d). Altogether, these data suggest that like canonical integrin ligands, the interactions of Tau and Tau PFFs with ITGαV/β1 also activate integrin signaling.

**Tau PFFs induce astrocytic inflammation via αV/β1 integrin.** As a major immune-competent cell in the CNS, astrocytes can sense danger signals and elicit an immune response via the secretion of pro-inflammation or anti-inflammation cytokines and chemokines[28,47]. We analyzed the mRNA expression of a collection of cytokine and chemokine genes by qRT-PCR in astrocytes exposed to Tau monomer or Tau PFFs using PBS-treated cells as a reference. Intriguingly, PFF treatment caused a strong induction of several pro-inflammatory cytokines (*IL1α*, *IL1β*, *IL6*, and *TNFα*) and chemokines (*Ccl2*, *Ccl3*, *Ccl4*, and *Cxcl10*). By contrast, the expression of *Clc12* and several anti-inflammation cytokines/chemokines such as *IL10* and *TGFβ* was not affected (Fig. 4a, b). Monomeric Tau also activated pro-inflammatory cytokines/chemokines, but consistently to lower levels (Fig. 4a, b), suggesting that integrin activation by Tau might be coupled to inflammation induction. Indeed, the induction of pro-inflammatory genes by Tau PFFs was largely abolished in integrin αV/β1 knockdown astrocytes (Fig. 4c). Thus, ITGαV/β1 is required for a Tau PFF-induced pro-inflammation response in astrocytes.

Astrocyte activation is usually linked to increased expression of the NO synthase (*NOS2*)[48], which upregulates the production of nitric oxide (NO) during inflammation[49]. Indeed, qRT-PCR detected a significant increase in the expression of *NOS2* in astrocytes following Tau PFF treatment. Like integrin target genes, *NOS2* expression was also induced by monomeric Tau, albeit by a smaller magnitude (Fig. 4d), and the depletion of *ItgαV/β1* significantly reduced Tau PFF-induced *NOS2* expression (Fig. 4e).

**Tau PFF-induced inflammation requires Talin1 and FAK.** To further elucidate the functional link between Tau PFF-induced integrin signaling and astrocytic inflammation, we used immunopurified astrocytes to test the requirement of Talin1 and FAK in PFF-induced inflammation. As expected, Tau PFF treatment induced the expression of pro-inflammation cytokines and chemokines such as *IL1α*, *IL1β*, *IL6*, *TNFα*, *Ccl2*, *Ccl3*, *Ccl4*, and *Cxcl10* as well as the *IL1* downstream gene *NOS2*, but noticeably, the induction of *IL1α*, *IL1β*, and *NOS2* by Tau PFF was much higher than that seen in conventionally purified astrocytes (Fig. 4f–i). This was probably due to the use of integrin antibodies during the purification (Fig. 3c), which might enrich a population of astrocytes with high integrin expression. As anticipated, lentivirus-mediated knockdown of *Talin1* reproducibly reduced Tau PFF-induced expression of pro-inflammation cytokines, chemokines and *NOS2* (Fig. 4f–i).

To test the role of FAK in Tau PFF-induced inflammation, we pre-treated immunopurified astrocytes with PF-562271, which dose-dependently abolished Tau PFF-induced expression of pro-inflammatory genes (Supplementary Fig. 4a and Fig. 4j, k). Thus, Tau PFF-induced inflammation requires Talin1 and FAK.

**Tau PFF activates NFκB via the integrin signaling.** Because NFκB is a central regulator of inflammation and because integrin activation has been linked to NFκB signaling[50,51], we used qRT-PCR to measure the expression of *NFκB* in PFF-treated and monomeric Tau-treated astrocytes. For comparison, we also tested *STAT3*, another inflammation regulatory transcription factor[52]. Indeed, *NFκB* but not *STAT3* was activated by Tau PFFs and also to a lesser extent by monomeric Tau (Fig. 5a). Additionally, PFF-induced NFκB expression was mitigated either by knockdown of *ItgαV/β1* or *Talin1*, or by the FAK inhibitor PF-562271 (Fig. 5b–d). Immunostaining using a NFκB specific antibody detected NFκB p65 mostly in the cytoplasm in control cells, but treatment with either monomeric Tau or Tau PFF caused an increase in the nuclear/cytoplasmic NFκB ratio (Fig. 5e–g), further confirming the activation of NFκB by Tau. These findings, together with the observations that Tau-induced inflammatory gene expression was abolished by a NFκB inhibitor PDTC (Pyrrolidine dithiocarbamate) (Supplementary Fig. 4b and Fig. 5h, i) suggest that Tau PFFs induce an inflammation response in astrocytes as a result of NFκB activation via integrin signaling.

**Tau PFF converts astrocytes into a neurotoxic state via αV/β1 integrin.** A recent study suggested that extracellular factors in the brain microenvironment can cause native astrocytes to adopt different functional states[30]. These states are linked to the expression of pan-reactive genes and certain A1-specific or A2-specific genes. Since Tau PFFs activate integrin signaling and

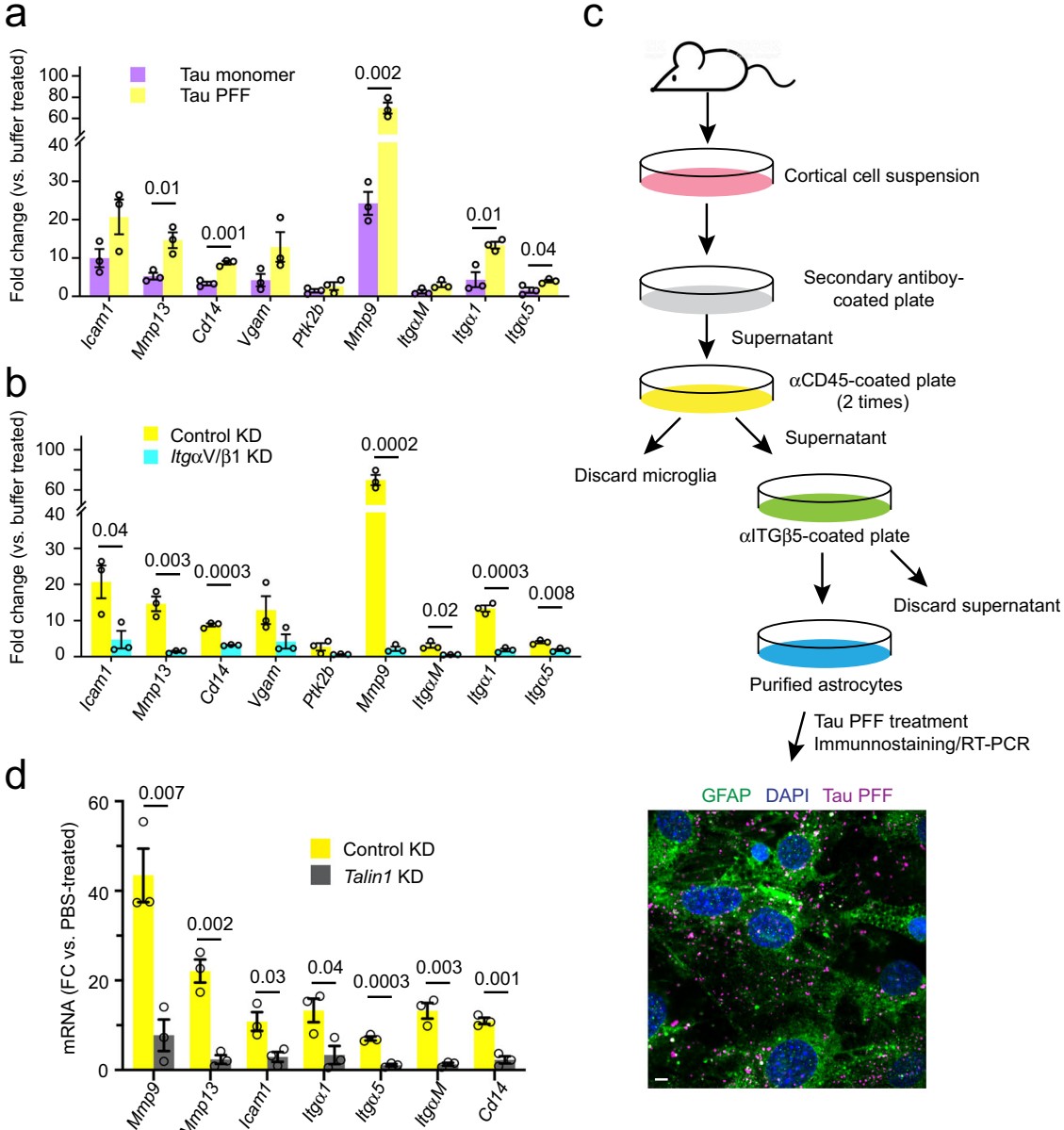

**Fig. 3 Monomeric and filamentous Tau differentially activate integrin signaling. a, b** Tau PFF activates integrin signaling in an ITGαV/β1 dependent manner. qRT-PCR analyses of the expression of a collection of integrin target genes in primary astrocytes. Cells were first infected with either control or ITGαV/β1 shRNA for 72 h and then treated with PBS, monomeric Tau, or Tau PFFs (200 nM) for 6 h. Shown are fold changes (FC) normalized to PBS-treated samples. **a** A comparison between Tau monomer and PFF. **b** A comparison of Tau PFF treatment in control vs. ITGαV/β1 knockdown cells. Mean ± SEM, n = 3 independent experiments, each with two technical repeats. **c** A schematic overview of the immuno-panning strategy for purification of astrocytes using anti-ITGβ5 antibodies. Briefly, cortical cell suspensions were passed successively over secondary antibody-coated and anti-CD45-coated plates to remove macrophages and microglia, and then plated in a positive selection panning plate for astrocytes (anti-ITGβ5 coated). The lower panel shows a representative immunofluorescence image of purified astrocytes treated with Tau PFF Alexa594 (200 nM) (Magenta) 1 h and then stained with anti-GFAP antibodies (Green) and Hoechst (Blue). Scale bar, 5 μm. **d** Talin1 is required for Tau PFF-induced integrin signaling. As in **b**, except that immunopurified astrocytes were first infected with either control- or Talin1 shRNA-expressing lentivirus and then treated with Tau PFF. Mean ± SEM n = 3 independent experiments, each with two technical repeats. The numbers and asterisks above indicate p value (****p < 0.0001) determined by two-tailed unpaired t-test.

induce inflammation, we tested whether PFF-treated astrocytes resemble the neurotoxic A1 state or the neuroprotective A2 state by determining the expression of the previously characterized pan-reactive, A1 and A2 signature genes. Interestingly, qRT-PCR showed that both Tau monomer and PFFs induced the expression of many pan-reactive and A1 specific genes such as *Gbp2, H2-T23, Ggta, Ligp1, Psmb8* etc with PFFs generally causing a higher level of induction than monomer (Fig. 6a). By contrast, most A2

genes were not significantly affected with the exception of *Cd14, Tgm1,* and *Ptx3.* Thus, monomeric Tau and Tau PFF appear to convert astrocytes into an A1-like state. Knockdown of either *ItgαV/β1* or *Talin1* dramatically reduced Tau PFF-associated gene induction (Fig. 6b, c and Fig. S5), so did the treatment with PF-562271 and the NF κB inhibitor PDTC[50] (Fig. 6d, e). Thus, Tau PFFs alter astrocytes in an integrin-dependent manner, converting them to an A1-like state.

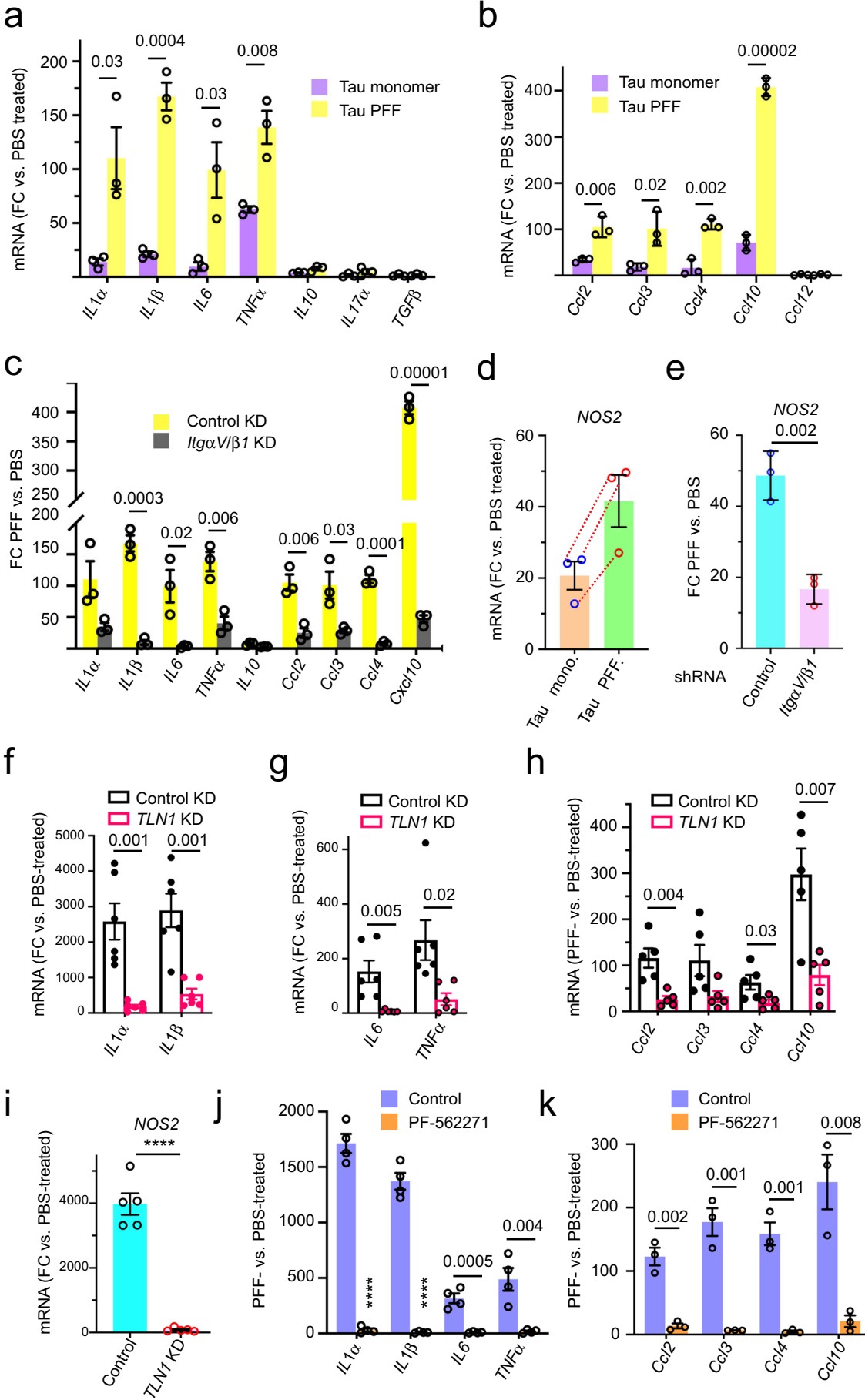

**Fig. 4 Tau PFF induces inflammation in astrocytes via the integrin pathway. a–c** Tau PFF induces astrocytic inflammation in an ITGαV/β1 dependent manner. qRT-PCR analyses of the expression of inflammation-associated cytokine and chemokine genes in primary astrocytes. Cells were first infected with either control shRNA (**a**, **b**) or control and ITGαV/β1 shRNA (**c**) for 72 h and then incubated with PBS, monomeric Tau or Tau PFFs (200 nM) for 6 h. Shown are fold changes (FC) normalized to PBS-treated cells. Mean ± SEM, $n = 3$ independent experiments, each with two technical repeats. **d**, **e** Tau PFF induces *NOS2* expression in an ITGαV/β1 dependent manner. **d** As in **a**, except that *NOS2* mRNA was analyzed. Mean ± SEM, $n = 3$. Dashed lines indicate individual experiment. **e** As in **c**, except that *NOS2* mRNA was analyzed. The number indicates $p$ value determined by two-tailed unpaired $t$-test. **f–h** qRT-PCR analyses of the expression of pro-inflammation-related cytokines (**f**, **g**) and chemokines (**h**) in immunopurified astrocytes. Cells were infected with either a control or Talin1 (TLN1)-specific shRNA prior to the treatment with either PBS or Tau PFFs (200 nM) for 6 h. Fold changes (FC) were normalized to PBS-treated samples. Mean ± SEM, $n = 6$ (**f**, **g**) or $n = 5$ (**h**) independent experiments, each with two technical duplicates. **i** As in **f**, except that *NOS2* mRNA was analyzed. Mean ± SEM, $n = 5$ independent experiments. **j**, **k** qRT-PCR analyses of the expression of the indicated pro-inflammation cytokines (**j**) or chemokines (**k**) in immunopurified astrocytes that were treated with DMSO or PF-562271 (1 µM) for 1 h before incubation with PBS or Tau PFF (200 nM) for 6 h. Fold changes (FC) were normalized to PBS-treated samples. Mean ± SEM, $n = 4$ (**j**) or 3 (**k**) independent experiments each with two technical duplicates. The numbers and asterisks above indicate $p$ value (****$p < 0.0001$) determined by two-tailed unpaired $t$-test.

**Tau PFF-treated astrocytes release neurotoxic factors**. To further test whether Tau PFFs converted astrocytes into a neurotoxic state, we treated astrocytes with PBS, monomeric Tau, or Tau PFFs for 6 h and then replated these cells in a Tau-free medium for 48 h. After incubation, many cells treated by Tau PFFs displayed a tadpole-like morphology, which was not seen in monomeric Tau or PBS-treated samples (Supplementary Fig. 6a). We then used the conditioned medium (CM) from these cells to treat mouse cortical neurons. No Tau carryover was detected in CM (Supplementary Fig. 6b). However, when neuronal cell death was monitored 72 h later by a fluorescence-based cell viability assay (Fig. 7a), the CM from Tau PFF-treated cells was significantly more toxic than that from PBS-treated or monomeric Tau-treated astrocytes. Interestingly, the CM from PFF-treated, ITGαV/β1 knockdown astrocytes had a significantly reduced neurotoxic activity compared to the CM from PFF-treated control cells (Fig. 7b, c). These results suggest that Tau PFF-treated astrocytes release a neurotoxic factor(s) in an ITGαV/β1-dependent manner.

## Discussion

The accumulation and propagation of Tau filamentous aggregates are observed in a broad spectrum of neurodegenerative disorders including AD[1,53]. The progressive spreading of Tau inclusions, which has been recapitulated in mouse models[4,5,54], is thought to take place along neuronal connections or via unconventional protein secretion followed by cell uptake[7,55]. Tau-containing aggregates have also been detected in glial cells (both microglia and astrocytes), although endogenous Tau is not expressed in these cells. This suggests that pathological Tau inclusions may also be transmitted from neurons to glia[56–59]. Importantly, expression of human Tau in glia in a *Drosophila* model led to neurotoxicity, suggesting that Tau, if propagated into glial cells, might have a pathogenic activity[60]. Likewise, in a transgenic mouse model, astrocytic expression of human Tau using the astrocyte-specific GFAP (glial fibrillary acidic protein) promoter leads to neurodegeneration[61]. These results demonstrate that the prion-like transmission of Tau aggregates from neurons to glia may contribute to neurodegeneration. However, it is unclear whether it is the internalized Tau fibrils or an interplay between the extracellular Tau and a cell surface receptor(s) that causes neurotoxicity.

In this study, we show that Tau filamentous aggregates bind to an integrin receptor complex (ITGαV/β1) on the astrocyte surface, which not only mediates Tau fibril uptake, but also regulates fibril-induced inflammation and astrocytic conversion (Fig. 7d). ITGαV/β1, as a member of the integrin family, is a cell surface adhesion molecule that binds to extracellular matrix proteins[44]. This activates integrin signaling, which has been linked to cancer cell invasion and proliferation[44,46]; Through the actin-binding protein Talin1 and the downstream focal adhesion kinase (FAK), integrin activation alters gene expression, allowing cancer cells to adapt to the constantly changing microenvironment during metastasis. Integrins and their ligands are also widely expressed in the CNS where they support neural development[62,63]. Moreover, several lines of evidence have hinted at a role for integrins in proteotoxic stress-associated neurodegeneration: 1) A genome-wide association study (GWAS) in a *Drosophila* model of AD identified fly homologs of the human integrin receptors ITGαM and ITGα9 as modifiers for Tau-induced neurotoxicity[36]; 2) Microglia can take up fibrillar AD-associated β-Amyloid (Aβ) aggregates via ITGα6/β1 and an integrin-associated protein CD47[39]; 3) The expression of β1 integrin is elevated in cortical astrocytes from AD patients, and in AD-related animal models β1 integrin can mediate soluble Aβ oligomers-induced astrogliosis[64]; 4) A more recent study revealed that in response to spinal cord injury, which is often associated with the release of stress-inducing factors such as misfolded proteins, astrocytes are activated via an integrin and cadherin-dependent mechanism to form astrocytic scars[65]. In light of these observations and our findings, we propose that integrin receptors on astrocytes may sense extracellular misfolded proteins, initially facilitating their clearance, but eliciting a danger signaling in the form of inflammation when overburdened by misfolded proteins. This model does not exclude the possibility that the astrocytic inflammation status may be partially modulated by endocytosed Tau-integrin complex, or by a combined activation of integrin and other cell signaling pathways on the cell surface.

Intriguingly, Tau fibrils and monomers can differentially activate integrin signaling, which might be due to their different oligomeric states. In support of this view, recent studies have underscored the role of liquid–liquid phase separation in transmembrane receptor signaling[66]. Specifically, the coagulation of membrane receptors into a gel-like state upon binding to a multivalent ligand can dramatically amplify the magnitude of downstream signaling. Indeed, receptor clustering upon binding of a multivalent ligand is known to potentiate the outside-in integrin signaling[67].

Astrocytes, as the most abundant glial cells in the CNS, are a well-established modulator of neurodegeneration-associated inflammation[47]. Specifically, disease-associated abnormal proteins such as Aβ and α-Syn fibrils can change astrocyte physiology. For instance, different Aβ species can activate disparate astrocytic cell receptors to induce the pro-inflammatory NF-κB pathway[68–71]. Our study shows that extracellular Tau aggregates can impact astrocytes similarly, suggesting a unified interplay between extracellular misfolded proteins and astrocytes.

Importantly, our study shows that Tau PFF-activated astrocytes resemble the neurotoxic A1 state reported recently[30]. This gain-of-function state, which has been associated with astrocytic

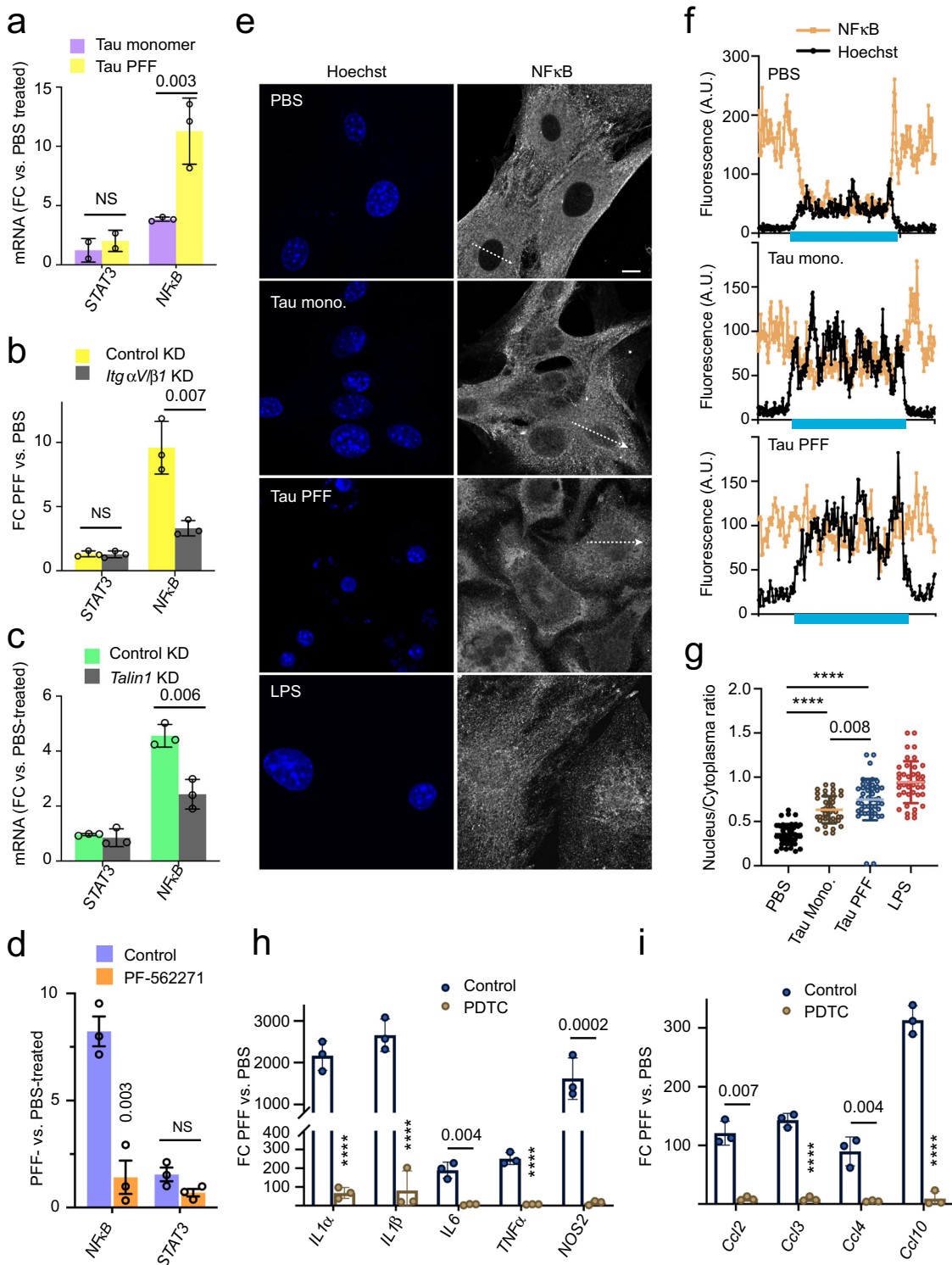

**Fig. 5 Tau PFF-induced inflammation is dependent on NFκB activation. a** qRT-PCR analyses of the expression of NFκB and STAT3 in primary astrocytes exposed to PBS, monomeric Tau or Tau PFFs (200 nM) for 6 h. Shown are fold changes (FC) normalized to PBS-treated cells. Mean ± SEM, $n = 3$ independent experiments each with two technical repeats. **b, c** As in **a**, except that cells were first infected with control or ITGαV/β1 shRNA (**b**), or Talin1 shRNA (**c**) for 72 h and then incubated with PBS or Tau PFFs (200 nM) for 6 h. Mean ± SEM, $n = 3$ independent experiments. **d** As in **b**, except that control-treated or PF-562271-treated astrocytes were used. Mean ± SEM, $n = 3$ independent experiments. **e–g** Tau PFF increases NFκB nuclear translocation. **e** Representative confocal images of astrocytes treated with either PBS, Tau monomer, PFFs (200 nM) for 4 h or LPS as a positive control (5 μM) for 30 min. Cell were fixed and stained with a NFκB antibody and Hoechst (blue). The graphs in **f** show the fluorescence intensity profile along the lines indicated in **e**. The blue bars indicate nuclear regions. **g** Quantification of the nuclear/cytoplasmic NFκB ratio in randomly selected cells. Mean ± SEM, $n = 3$ biologically independent experiments. **h, i** Tau PFF-induced inflammation is suppressed by a NFκB inhibitor. As in **d**, except that cells were pre-treated with PDTC (90 μM) for 1 h before Tau PFF and PBS treatment and that the indicated genes were analyzed. Mean ± SEM, $n = 3$ independent experiments. The numbers and asterisks above indicate $p$ value (****$p < 0.0001$) determined by two-tailed unpaired $t$-test.

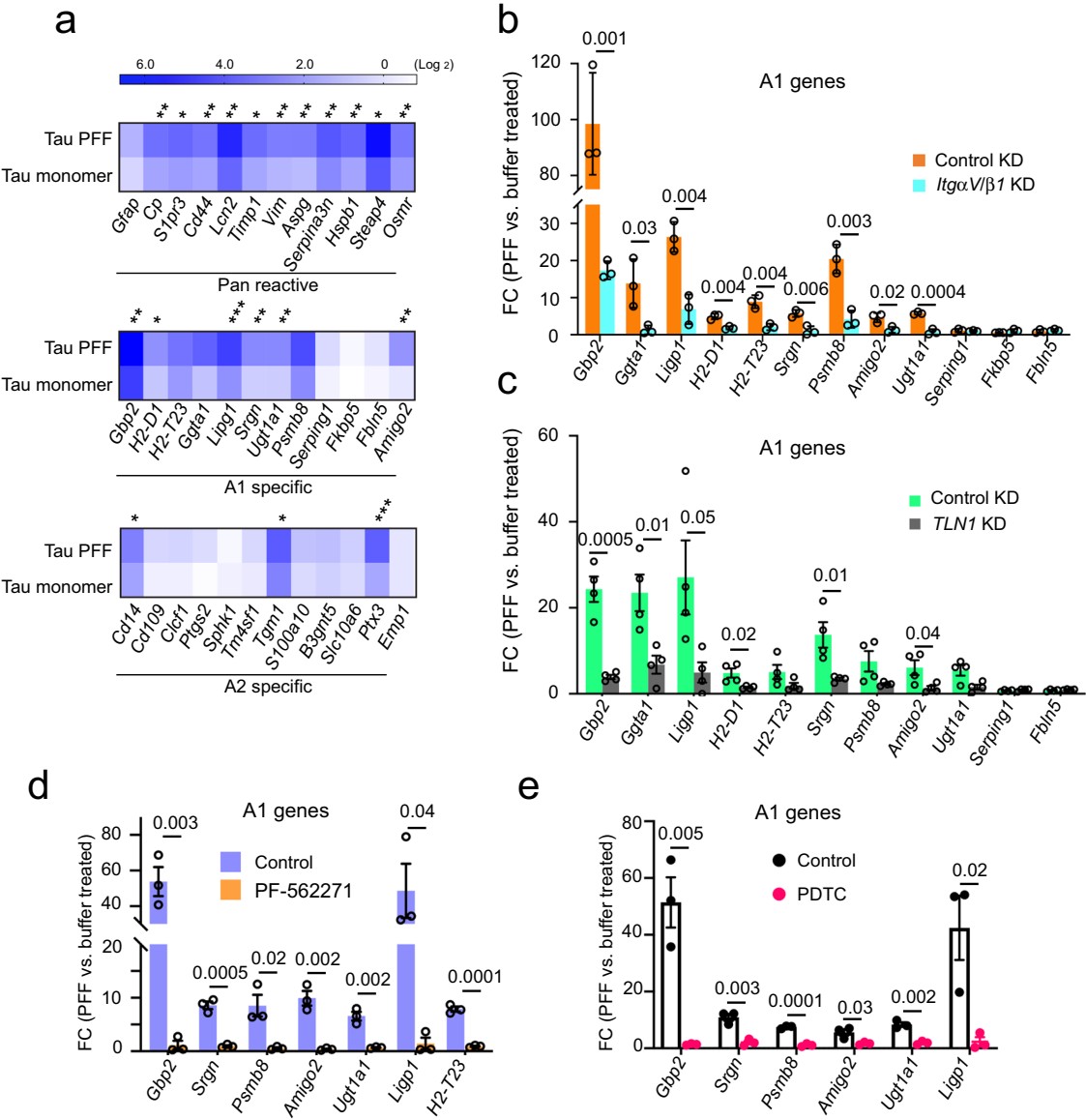

**Fig. 6 Tau PFF converts astrocytes to a neurotoxic state in an integrin-dependent manner. a** A heat map shows the expression of pan-reactive, A1 and A2 signature genes in Tau monomer- and PFF-treated primary astrocytes (200 nM, 6 h), which is normalized to PBS-treated cells. Shown is the average of three independent experiments. Asterisks indicate *p* value (\**p* < 0.05; \*\**p* < 0.01; \*\*\**p* < 0.001) from two-tailed unpaired *t*-test. **b** The expression of the indicated A1 genes were analyzed by qRT-PCR from PBS- or Tau PFF-treated (200 nM, 6 h) astrocytes, which had been infected with lentiviruses (72 h) expressing control or ITGαV/β1 shRNAs. Fold changes (FC) were normalized to PBS-treated samples. Mean ± SEM, *n* = 3 independent experiments each with two technical duplicates. **c** As in **b**, except that immunopurified astrocytes infected with control-expressing or Talin1 (TLN1) shRNA-expressing lentiviruses were used. Mean ± SEM, *n* = 4 biologically independent samples. **d** As in **c**, except that cells were treated with DMSO or PF-562271 (1 μM, 1 h) instead of lentivirus prior to PBS or Tau PFF treatment. Mean ± SEM, *n* = 3 biologically independent samples, each with two technical duplicates. **e** Tau PFF-induced A1 genes is suppressed by a NFκB inhibitor. As in **d**, except that cells were treated with PDTC (90 μM) for 1 h before Tau PFF and PBS treatment. Mean ± SEM, *n* = 3 biologically independent samples. The numbers above indicate *p* value determined by two-tailed unpaired *t*-test.

inflammation, was suggested to play a causal role in neurode-generation, but the upstream signals that promote astrocytic activation are unclear. Particularly, existing evidence suggests that the A1 state could only be achieved by a relayed signal elicited from microglial cells exposed to an extracellular insult such as LPS. By contrast, we showed that even in the absence of micro-glia, misfolded Tau aggregates can trigger astrocyte activation via ITGαV/β1. This suggests that astrocytic inflammation may be a key contributor to tauopathies. Additionally, given that neurons are usually the major source of secreted Tau, our findings raise the possibility that astrocytes may monitor the healthy state of neurons by sensing the level of misfolded Tau released. It is

conceivable that Tauopathies in neurons at low levels may be mitigated by intercellular transmission and astrocyte-mediated clearance, but at high levels will activate astrocytes, which release neurotoxic factors to cause neurodegeneration. This model opens a venue for drug development aimed at preventing astrocytic activation in tauopathies.

## Methods

**Animals and reagents**. Wild-type C57BL/6J mice were purchased from the Jackson Laboratories. All animals were maintained in accordance with the animal care standards of the National Institutes of Health. All reagents are listed in Supplementary Table 1. Animal studies were conducted following the animal study

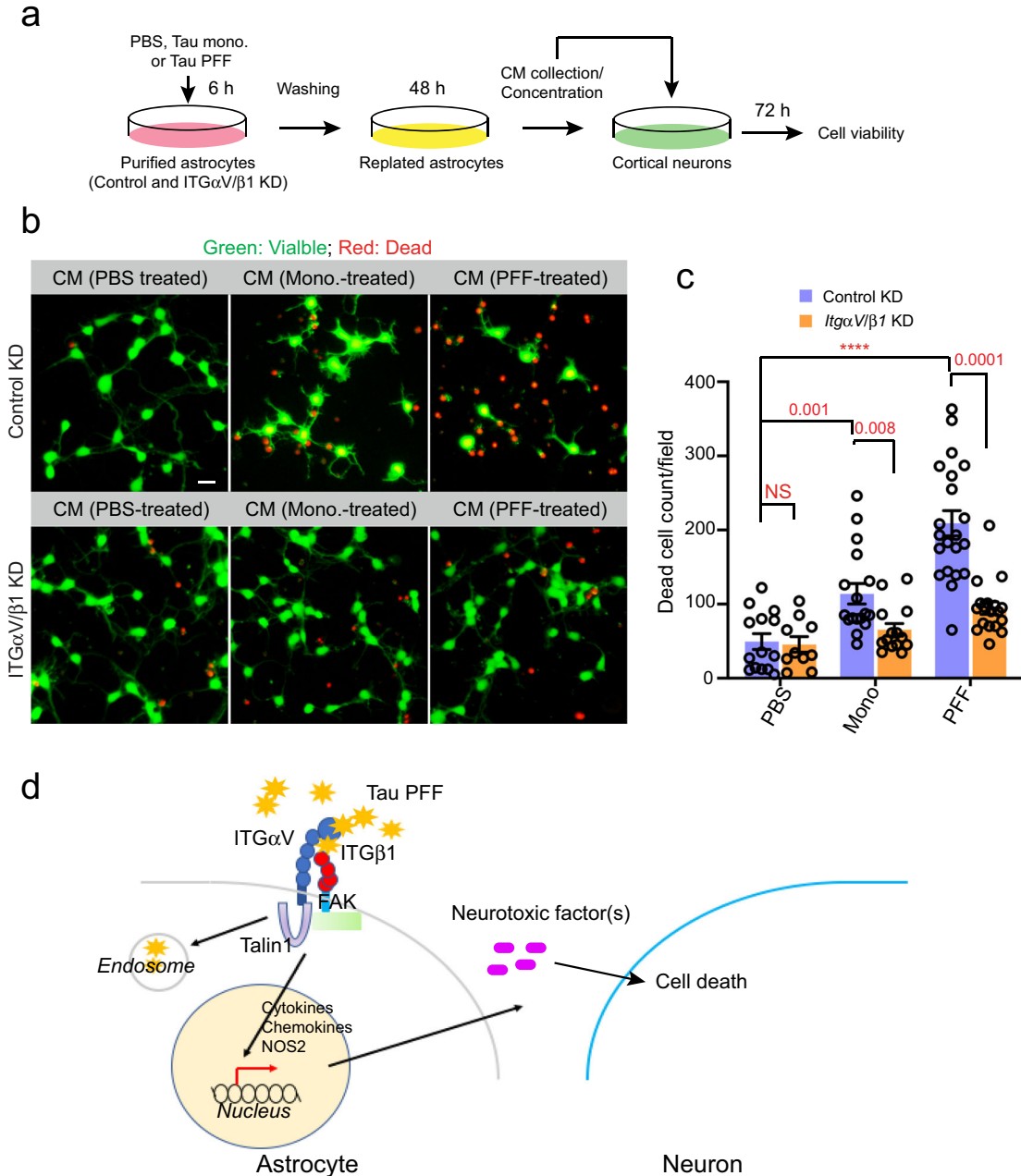

**Fig. 7 Tau PFF-treated astrocytes release a neurotoxic factor(s) via ITGαV/β1 activation. a** Scheme of the cell toxicity assay. **b, c** Primary cortical neurons at DIV 10 were treated with the indicated condition medium (CM) for 72 h and stained with a green-fluorescent dye calcein-AM to label viable cells and a red-fluorescent dye ethidium homodimer-1 to label dead cells. **b** Representative images. Scale bar, 10 μm. **c** Quantification of dead cells labeled by ethidium homodimer-1 in randomly selected fields. Mean ± SEM, n = 3 biologically independent samples. The numbers and asterisks above indicate p value (****p < 0.0001) determined by two-tailed unpaired t-test. **d** A working model showing the interplays between Tau-PFF, astrocytes and neurons.

protocol ASP K117-LMB-17 approved by the NIDDK Animal Care and Use committee chaired by Dr. Constance Noguchi.

**Conventional primary astrocyte culture.** Primary astrocyte cultures were prepared from cerebral cortices of P2-P5 C57BL/6J mice. Cortices were dissected, stripped of meninges, and digested with 0.25% trypsin at 37 °C in Hank's Balanced Salt Solution (HBSS) (Thermo Fisher) for 10 min. Trypsinization was stopped by the addition of astrocyte culture medium (DMEM/ F12 50/50 (Thermo Fisher) containing 25 mM glucose, 4 mM glutamine, 1 mM sodium pyruvate, and 10% FBS). Single-cell suspension of the digested tissue was obtained by repeated pipetting. Cells were seeded into a 75 ml flask at a density of ~4 × 10⁵ cells/cm² and cultured in astrocyte culture medium at 37 °C in a humidified 5% CO₂ incubator. Monolayers of glial cells were obtained 7–10 days after plating. To remove microglia, cultures were gently shaken, and the floating cells (microglia) were removed, resulting in more than 95% pure astrocytes. The remaining astrocytes

were incubated for 72 h before being infected with lentivirus or other experiments. Before experiments, astrocytes were dissociated by trypsinization and then reseeded at $4 \times 10^5$ cells cm² or $1.5 \times 10^5$ cells per well in 24-well or $1.5 \times 10^6$ cells per well in 6-well or $2 \times 10^5$ cells cm² in Labtek imaging dish in DMEM F12 50/50 containing 10% FBS and 1% penicillin-streptomycin. At the start of each experiment, the medium was changed to regular DMEM without serum and antibiotics.

**Primary astrocyte culture purified by immuno-panning.** Astrocytes were purified by immune-panning from P2-P5 C57BL/6J mice and cultured as previously described[72]. Briefly, cortices were digested by Trypsin at 37 °C and then mechanically dissociated to generate single-cell suspension, which was incubated in successive negative immune-panning plates to remove macrophage and microglia cells before positive selection for astrocytes by an ITGβ5-coated panning plate. Isolated astrocytes were cultured in a defined, serum-free base medium containing 50% neurobasal, 50% DMEM/F12, 100 U/ml penicillin, 100 μg/ml streptomycin,

1 mM sodium pyruvate, 292 μg/ml L-glutamine, 1 × G5 supplement. This medium was supplemented with the astrocyte-required survival factor HBEGF (Peprotech, 100-47) at 5 ng/ml as previously described[73].

**Primary neuron cultures**. Primary neuron cultures were either purchased from Thermo Fisher Scientific or prepared from cerebral cortices of P0-P1 C57BL/6 mice. Cortices were dissected, stripped of meninges, and digested with 0.25% trypsin at 37 °C in HBSS for 10 min. Trypsinization was stopped by the addition of FBS. A single-cell suspension of the digested tissue was obtained by repeated pipetting. Cells were seeded into a 24-well plate at a density of 1.5 ×10⁵ cells per well and cultured in DMEM culture medium supplemented with 10% fetal bovine serum (Thermo Fisher), 100 U/ml of penicillin, 0.1 mg/ml of streptomycin (Thermo Fisher), 25 mM glucose, 4 mM glutamine, and 1 mM sodium pyruvate at 37 °C. At 24 h after seeding, the medium was changed to Neurobasal medium (Thermo Fisher) supplemented with B-27 (Thermo Fisher) containing 0.5 mM glutamine. Cells were cultured at 37 °C in a humidified chamber supplemented with 5% $CO_2$. Cultures were used for experiments from 7 to 10 days after seeding.

**Purification of Tau protein**. The recombinant human (Tau 2N4R isoform), containing 2 N-terminal inserts and four microtubule binding repeats was isolated as previously described[31]. Briefly, the untagged Tau protein was expressed using the pET29b vector in the *E.coli* strain *BL21*. From a culture volume of 10 l, the cell pellet was resuspended in ice cold cell resuspension buffer (20 mM MES, 1 mM EGTA, 0.2 mM $MgCl_2$, 5 mM DTT, 1 mM PMSF). Cells were disrupted by sonication, and NaCl was added to a final concentration of 500 mM. Cell suspension was boiled for 20 min, which denatured almost all proteins except for Tau. The denatured proteins and insoluble cell debris were sedimented by centrifugation at 127,000 × g for 40 min at 4 °C. The supernatant was dialyzed overnight twice in the cation exchange chromatography buffer A (20 mM MES, 50 mM NaCl, 1 mM EGTA, 1 mM $MgCl_2$ with 2 mM DTT, 0.1 mM PMSF at 4° under constant stirring. Cleared the dialysate by centrifugation at 127,000 × g for 40 min at 4 °C. The clear supernatant was loaded onto a cation-exchange chromatography column. The bound Tau protein was eluted with a linear gradient of 0–60% final concentration of the cation exchange chromatography buffer B (20 mM MES, 1 M NaCl, 1 mM EGTA, 1 mM $MgCl_2$, 2 mM DTT, 0.1 mM PMSF) over six column volumes. Fractions containing Tau were pooled and concentrated by an ultrafiltration device (Millipore 10 kDa MW cutoff) to a final volume of 1 ml. Concentrated Tau protein was further fractionated by a size exclusion column in a gel filtration buffer (PBS with 1 mM DTT), flash-frozen in liquid nitrogen and stored at −80 °C. Fifty micromolar purified Tau was labeled with Alexa Fluor 594 dye following the manufacturer's instruction (Thermo Fisher).

**Preparation of paired helical filaments from Tau proteins**. The condition for Tau polymerization is: 50 μM purified Tau protein, 12.5 μM heparin, a protease inhibitor cocktail (10 μg/ml leupeptin, 5 μg/ml chymostatin, 3 μg/ml elastatinal, and 1 μg/ml pepstatin), and 2 mM DTT in phosphate buffer saline (PBS). This solution was incubated at 37 °C for two weeks. Tau polymerization is induced by the addition of heparin. One millimolar of fresh DTT was added into the solution every 24 h to maintain reducing condition since disulfide bond formation in the four-repeat domain of Tau reduces polymerization efficiency. After polymerization, Tau filaments were dialyzed extensively with PBS to remove heparin and DTT. Transmission electron microscopy was used to check PHF assembly.

**Electron microscopy (EM) analysis of Tau filaments**. 5 mg/ml Tau filaments were diluted 10-fold in PBS and loaded onto an EM grid. The grid was then washed with pure water and stained with 3% uranyl acetate for 1 min, and then imaged with a Morgagni 268 transmission electron microscope.

**Tau internalization assay**. Tau fibrils were diluted in PBS and sonicated with 10 pulses of 30% amplitude before being added to culture medium. Primary astrocyte culture was treated with 200 nM Tau filament-Alexa Fluor 594 for 2 h in the astrocyte culture medium. In the case of inhibitor treatment, culture was pretreated with 10 μM PF262271 for 2 h. Cells were either directly imaged using a Zeiss LSM780 confocal microscope or fixed with 4% paraformaldehyde in PBS before imaging.

**RNA isolation and gene expression analysis**. RNA isolation was performed using a RNeasy Mini Kit (Qiagen) according to the manufacturer's protocol. RNA concentration was measured using Nanodrop-1000. Complementary DNA synthesis was performed using an iScript™ Reverse Transcription Supermix for RT-qPCR (Bio-Rad) according to the manufacturer's instructions, with a minimal input of 200 ng total RNA. Quantitative PCR (qPCR) was performed using a CFX96 Touch Real-Time PCR Detection System (Bio-Rad) using cDNA amount equivalent to 1–2 ng total RNA during cDNA synthesis. SsoAdvanced Universal SYBR Green Supermix (Bio-Rad) and a 2 pmol/ml mixture of forward and reverse primers were used for 45 cycles of gene amplification. The primers used for qPCR are listed in Supplementary Table 2. GAPDH mRNA was used as an internal reference.

The CFX manager software was used to analyze the qPCR results. To calculate fold changes, we used PBS-treated samples as a reference.

**Lentivirus production and infection**. For Lentivirus production, two 15 cm dishes of HEK293FT cells were seeded at 40% confluence. On the next day, 1 h prior to transfection, the medium was replaced with 13 ml pre-warmed Opti-MEM medium (Thermo Fisher). Transfection was performed using Lipofectamine 2000 and the PLUS reagent (Thermo Fisher). For each dish, 6.8 μg pCMV-VSV-G, 10.1 μg psPAX2 (Addgene), 13 μg gene-specific lentiviral shRNA plasmids and 135 μl of PLUS reagent (Thermo Fisher) were added to 4 ml Opti-MEM as mixture A, which is then mixed with mixture B containing 68 μl lipofectamine 2000 and 4 ml Opti-MEM. The complete mixture was incubated for 20 min at room temperature and then added to cells. After 6 h, the medium was changed to 25 ml D10 medium (DMEM medium with 10% FBS and 1% Bovine Serum Albumin) with antibiotics (penicillin/streptomycin, 10 U/ml) for virus production. After 60 h of incubation, virus-containing medium from two culture dishes were combined and centrifuged at 2000 × g at 4 °C for 10 min to pellet cell debris. The supernatant was filtered through a 0.45 μm low protein-binding membrane (Steriflip HV/PVDF, Millipore). To concentrate lentivirus, the cleared supernatant was ultracentrifuged at 47,000 × g for 2 h at 4 °C using the JA25.50 rotor (Beckman). The virus was resuspended overnight in 180 μl D10 medium at 4 °C. Virus was aliquoted, flash-frozen in liquid nitrogen and stored at −80 °C. For infection, 10 μl concentrated virus was directly added to astrocytes cultured in 6-well. The medium was changed 24 h after infection, and the knockdown efficiency was evaluated by qRT-PCR 96 h of postinfection.

**Cell lines and DNA transfection**. HEK293T and SH-SY5Y cells were purchased from ATCC. HEK293FT cells were from Thermo Fisher. HEK293T and HEK293FT Cells were maintained in Dulbecco's Modified Eagle's Medium (DMEM, Corning) containing 10% fetal bovine serum (FBS) and antibiotics (penicillin/streptomycin, 10 U/ml). SH-SY5Y cells were maintained in DMEM/F12 50/50 (Thermo Fisher) containing 10% fetal bovine serum (FBS) and antibiotics (penicillin/streptomycin, 10 U/ml). All cell lines were maintained at 37 °C. Cell transfections were performed using TransIT-293 reagent (Mirus) for HEK293T or Lipofectamine 2000 (Thermo Fisher) for SH-SY5Y cells following the manufacturer's instructions.

**Immunoblotting**. Immunoblotting was performed using a standard protocol. Proteins were separated in NuPAGE (4–12%) Bis–Tris gels (Thermo Fisher) and transferred onto nitrocellulose membranes (Bio-Rad). The target protein was detected by specific primary antibodies followed by secondary horseradish per-oxidase (HRP)-conjugated antibodies (for less abundant proteins) (Sigma) or fluorescence-labeled secondary antibodies (for abundant antigens) (Thermo Fisher). For immunoblotting with HRP-conjugated secondary antibodies, the signal was detected by the enhanced chemiluminescence method (ECL) using the Immobilon western chemiluminescent HRP substrate (Millipore), and recorded by a Fuji LAS-4000 image reader. The intensity of the detected protein bands was quantified by ImageGauge v3.0 software. For immunoblotting with fluorescence-labeled secondary antibodies, membranes were scanned using a LI-COR Odyssey scanner. The intensity of the protein bands was quantified by the LI-COR Odyssey v3.1 software.

**Cell surface binding assay and biotinylated samples preparation for MS proteomics**. Cells were washed with pre-cold PBS two times to remove serum completely. We prepared 200 nM Tau-Apex2 or Apex2 proteins in serum-free pre-cold DMEM/F-12 medium (5 ml for each 15 cm petri-dish). We incubated cells with these proteins for 30 min at 4 °C with gentle shaking. After incubation, we removed the unbounded proteins by thoroughly washing the cells three times, each with 6 ml pre-cold F12 medium. We then incubated cells in 500 μM Biotin-phenol (BP) in F-12 serum-free medium (10 ml for one 15 cm petri-dish) pre-warmed to 37 °C. After 30 min of incubation, we added freshly prepared $H_2O_2$ stock solution (100 mM) into the cells to achieve a final concentration of 1 mM. Cells were incubated further at room temperature for 4 min. The medium was removed and cells were washed twice with the quencher solution (10 mM sodium ascorbate, 5 mM Trolox, and 10 mM sodium azide in DPBS.), followed by two washes with DPBS, and then resuspended in the quencher solution. Collect Cells were pelleted by centrifugation at 3000 × g for 10 min at 4 °C, lysed in 7 ml PBS-based lysis buffer with 0.5% NP40, a protease inhibitor cocktail, 1 mM DTT, 10 mM sodium azide, 10 mM sodium ascorbate and 5 mM Trolox). After incubation on ice for 40 min, cell extracts were cleared by centrifugation at 15,000 × g for 10 min at 4 °C. Cleared supernatant was incubated with 50 μl pre-washed streptavidin beads, gently rocked at 4 °C for 1 h. The beads were extensively washed with PBS containing 0.05% NP40. Bound proteins were eluted in 50 μl 1× sample buffer. Proteins were fractionated on a SDS-PAGE gel. Proteins in excised bands were identified by the Taplin Mass Spectrometry core at Harvard Medical School as fee for service.

**Survival/cell toxicity assay**. Astrocytes were infected with shRNA-expressing lentivirus on 72 h after plating. Cells were grown for a total of 7 days in serum-free medium supplemented with 5 ng/ml HBEGF18 in 6-well plate. Cells were then

treated with 200 nM Tau PFFs or an equivalent volume of DPBS (as a control) for 6 h, then washed extensively to remove remaining Tau filaments. To further ensure that no Tau filaments were carried over to conditioned medium, cells were trypsinized and then re-seeded into a new 6-well plate, grown for an additional 48 h. At this time, the conditioned medium was collected and a protease inhibitor cocktail was added to inhibit protein degradation. Conditioned medium was concentrated ~40 times by Amicon Ultra-15 Centrifugal Filter Units with 30 kDa cutoff filter (Millipore, UFC903024), and then added to primary neuron cultures at 10 days in vitro (plated at $1.5 \times 10^5$ cells per well in a poly-D-lysine-coated 24-well plate). The viability assay was performed 72 h later using the Live/Dead Kit (Thermo Fisher Scientific, L3224).

**Image acquisition and processing**. Confocal microscopy was performed on a Zeiss LSM780 laser scanning confocal microscope (63×/1.46 objective) with the Zen program. Image processing was done using the open source ImageJ (Fiji) software. Images were converted to individual channels and regions of interest were drawn for intensity measurement. For dead cell counting, single channel images were converted to black and white. A constant threshold was applied to all the images for dot counting.

**Statistics and reproducibility**. Statistical analyses were performed using either Excel or Prism 8. Data are presented as mean ± SEM, calculated by GraphPad Prism 8, and p-values were calculated by student's t-test using Excel. The n values in the figure legends indicate independent experimental repeats. Images were prepared with Adobe Photoshop and graphs were plotted by GraphPad Prism 8.

Figure 1c is a representative gel from two independent biotin pulldown experiments. Figures 1d–f, 2a and Supplementary Figs. 2a, b, e, 3a show representative gels from three independent experiments. Supplementary Figs. 1c, 2c, 3b, c, 6a, b show negative controls that were only performed once.

**Reporting summary**. Further information on research design is available in the Nature Research Reporting Summary linked to this article.

## Data availability

All data generated for this study are included in this article and its supplementary materials. Source data is available with this paper. All additional information is available from the corresponding author upon reasonable request. Source data are provided with this paper.

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

## Acknowledgements
We thank R. Tycko (NIDDK) for assistance with the negative stain EM, J. Reece (NIDDK) for imaging assistance, Q. Zhang and Y. Xu (NIDDK) for critical reading of the manuscript. P. Wang, and Y. Ye are supported by an intramural research program of NIDDK.

## Author contributions
P. Wang, and Y. Ye designed the research, analyzed the data. P. Wang performed the experiments. P. Wang and Y. Ye wrote the paper together.

## Competing interests
The authors declare no competing interests.
