## [Peer Review File · Nature Communications]

Reviewers' Comments:

Reviewer #1:

Remarks to the Author:

In this interesting paper, astrocytic ITGaV/b1 receptors are identified as binding partners for tau that allow tau internalisation. The resulting activation of ITGaV/b1 signalling initiates inflammatory responses in astrocytes that confer neurotoxicity. I have several major concerns, that should they be addressed would produce a compelling paper:

1. The sample size of most experiments is very small (n=3) and sometimes insufficient (n=2). There appears to be considerable inter-experiment variation (particularly notable in Fig. 6). The sample size should be increased, and n should be included for each panel in the figure legends. What accounts for the large variation in readings found for control conditions in Fig. 6 A-C?
2. There are some incorrect statements in the introduction and discussion about tauopathies including Page 3 "intracellular neurofilaments (IFLs)" are described as a pathological hallmark of neurodegenerative diseases that are formed by hyperphosphorylated tau. Neurofibrillary tangles (NFTs) are the hallmark formed by hyperphosphorylated tau in AD. Neurofilaments may be associated with these lesions but are not their primary component. Similarly on Page 14, line 279 "tauopathies" are not found within glial cells, rather tau aggregates or neurofibrillary pathology accumulates in glia in some tauopathies.
3. The experiments rely on use of tagged recombinant tau. the relevance of heparin-induced tau filaments has been questioned due to differences with brain-derived tau (Zhang et al., 2019 DOI: 10.7554/eLife.43584). Confidence would be improved if key findings were replicated using filamentous and monomeric tau extracted from postmortem human brain or from transgenic mouse models of tauopathy.
4. The effect on astrocyte viability and proliferation in response to tau monomer and PFF, shRNA and kinase inhibitors, particularly for experiments with prolonged incubation times (including Fig. 3-6 and presumably 7) should be examined. This is critical for the proper interpretation of the results gained.
5. The effects of cell incubation on ice for 2/3 hours on cell viability/membrane integrity should be shown (Figs S2, 2). Why were some incubations for 3 hours (Fig. S2A) and others for 2 hours (Fig. S2B, Fig. 2)?
6. Fig. 1. Since monomeric tau is shown to bind to ITGaV/b1, can an interaction of endogenous tau with endogenous ITGaV/b1 be demonstrated in astrocytes?
7. Fig. 4. The NFkappaB findings would be strengthened by analysis of NFkappaB and phospho-NFkappaB protein amounts, in addition to effects of treatment on NFkappaB translocation to the nucleus.
8. Fig 6C-D. It would be interesting to see if A2 transcript changes are reciprocal to those found for the A1 transcripts. This would help determine if astrocyte phenotype is recovered in these conditions.
9. Fig. 7. The viability and numbers/density of treated and replated astrocytes should be assessed. It would be useful to include a cytoskeletal marker in 7B to show the outline of neurons.

Minor issues:

1. Page 6. Cite previous references showing that tau primarily binds to the plasma membrane via its N-terminus (e.g. Brandt et al., 1995) and Pooler et al., 2012).
2. Fig. 1. Define WEC in the legend.
3. Fig. 1F legend. Add the concentration of Tau-APEX2 used, and the time of incubation with cells, to the legend.
4. Fig. 2B. Correct typos in figure headings (IGT).
5. Page 9, line 180. Add references to support the statement that "...fibronectin....downstream genes."
6. Page 9, line 183. Correct typo "phosphate-buffered saline" not "phospho-buffered"
7. Fig 3D: define what is meant by experimental pairs. Is n=2 here?
8. S. Figure 3. Why does recombinant tau yield bands lower than its predicted molecular weight

(75kDa) in these IPs?

9. Fig. 7. Add info about the age (DIV) or primary neurons to the figure legend and the amount of CM added.

10. Discussion, Pg 14, line 283. Reference 57 does not study propagated tau, only expression of tau in astrocytes. Please correct the wording of this sentence.

Reviewer #2:

Remarks to the Author:

In this manuscript by Wang & Ye, they provide a mechanism by which Tau in both monomeric and pre-formed fibrillar form can drive a reactive state in astrocytes. The authors promote the hypothesis that this reactive state is similar to the recently described neurotoxic/A1 inflammation-induced reactive astrocyte subtype, and further show that secreted from these Tau-mediated reactive astrocytes is toxic to neuron cultures. Data to support their model shows involvement of α V/B1 integrin as a receptor for Tau on the surface of astrocytes, and downstream activation pathways including NfKB and FAK-associated inflammation pathways. The authors are to be applauded for using cell culture methods that exclude serum in a number of experiments - as this is particularly important for studying immune and glial cell function.

Overall this is a novel pathway that differs from the original inflammation-microglia-astrocyte-neurotoxin axis that was described, and provides some interesting insights into the possibility for multiple mediators of a common reactive state.

I do have a few concerns with this study that would benefit from clarification:

1. production of 'A1'/neurotoxic reactive astrocytes using Tau. This unfortunately was the weakest section of the manuscript. There is no doubt that Tau (either in monomeric or fibrillar form) is driving transcriptomic changes in astrocytes, however whether this is the same as the previously published neurotoxic subtype remains unclear. The original description showed alteration in several cassettes of genes (A1/A2 as described here, but also 'PAN' reactive genes that were common to several reactive subtypes - e.g. Gfap, Lcn2, among others). The smaller subset of genes shown here may, or may not, show a faithful recapitulation of this response (indeed if they do not, but instead promote a different reactive state with an equally toxic function - this would still be very interesting). This could be addressed by analysis of PAN reactive genes from the original publication (REF 30 in this current manuscript), or by unbiased RNASeq to determine other important transcriptomic changes.

This is also true for the very interesting result (around line 259) that shows the addition of PF-5622 can decrease the upregulation of many reactive astrocytes genes. It would like to see it explored further - actually looking at all the gene expression changes in Tau-treated astrocytes, and how the PF-562271 changes this.

2. neurotoxic capacity. While it is apparent that PFF>monomeric>no Tau astrocyte conditioned media is able to kill neurons in a culture dish, there is no Tau only controls included in these studies. As the authors comment on line 48/49, filamentous Tau itself has been shown to be toxic to neurons - this should be removed from the conditioned media before completing these studies, or compared to a set of Tau only controls.

3. the authors have not considered an internally-regulated function of Tau (by astrocyte phagocytosis) that could mediate some form of reactive response. Can the authors comment on this, and is it possible to study the effects of Tau binding only to α V/B1 integrin by blocking astrocyte phagocytosis?

Minor concerns:

1. line 75 - there is also evidence that inflammation-induced astrocyte reactive states have decreased phagocytic capacity

2. line 82 - the original description of A1/neurotoxic astrocytes (REF 30) specifically highlighted that they did not lose neuro-supportive functions, but that they gained a specific neurotoxic capacity that overcame this support.
3. The authors note (paragraph starting at line 190) that traditional serum-containing cultures of astrocytes often have many contaminating cells, hence their validation of results using immunopanned astrocytes. The purity of these cultures would also be nice to know.
4. there are a small number of grammatical/sentence structure errors in the manuscript (e.g. line 64 'on the uptake side')
5. Figures - please label micrographs with antibodies/fluorescent channels being imaged (e.g. Fig 2f/h) and add missing scale bars (e.g. Fig 3)
6. as per convention, gene names should be italicized (in text, and figures)

Reviewer #3:

Remarks to the Author:

The authors use a proximity ligation assay to identify a receptor for tau on primary astrocytes *in vitro*, identifying the integrin receptor complex (ITGA ν / β 1). The authors also find that the receptor signals through Talin and FAK, leading to A1-like astrocyte activation and downstream neuron death. This finding is significant because it identifies a mechanism by which astrocytes internalize misfolded tau protein and could thus contribute to pathology during tauopathy. This has the potential to influence tauopathy/neurodegeneration fields, but also represents a novel astrocyte-neuron communication mechanism since tau release increases with neuron firing. The study appears to use appropriate statistical tests and includes methods that would allow for other researchers to reproduce the work.

Overall the work well done but I do have some concerns:

1. How did you ensure that the recombinant tau was endotoxin-free? As mentioned previously LPS activated astrocytes increase phagocytosis and cytokine/chemokine release etc. This is essential to validate all of the results from the paper. It could be helpful to do an experiment with human or mouse-derived PFFs or seeds isolated, for example, by size exclusion chromatography, to ensure this is not an effect of using E Coli to produce the tau.
2. The authors suggest that tau-integrin signaling acts via STAT3 and NF κ B, but the methods used to assert this are weak. Expression levels of nf κ B or STAT3 mRNA (as used here) is not really a readout of NF κ B activation. I recommend looking at I κ B α degradation and/or phospho-p65 accumulation and/or nuclear translocation by Western blot. Phospho STAT3 Western blot also would be helpful. Strongest evidence could be through luciferase reporter assay for these TFs with astrocyte PFF treatment.
3. The authors suggest that tau-integrin signaling leads to A1 astrocyte activation. It would be helpful to see a positive control of known A1 inducers to compare the effect of tau monomers and PFFs.

We thank the reviewers for the positive comments and constructive suggestions.

Reviewer #1 (Remarks to the Author):

In this interesting paper, astrocytic ITGaV/b1 receptors are identified as binding partners for tau that allow tau internalisation. The resulting activation of ITGaV/b1 signalling initiates inflammatory responses in astrocytes that confer neurotoxicity. I have several major concerns, that should they be addressed would produce a compelling paper:

1. The sample size of most experiments is very small (n=3) and sometimes insufficient (n=2). There appears to be considerable inter-experiment variation (particularly notable in Fig. 6). The sample size should be increased, and n should be included for each panel in the figure legends. What accounts for the large variation in readings found for control conditions in Fig. 6 A-C?

RE: We have repeated a few key experiments to increase the sample size. Specifically, all the qRT-PCR experiments are done with at least 3 independent batches of primary astrocytes (some with more). Please note that the n value specifies **independent preparation of astrocytes as opposed to mouse number**. Each preparation was usually done with 4-6 pups of mixed genders.

The inter-experiment variation is a reflection of the fact that primary astrocytes are used (compared to stable cell lines derived from clonal cells). The experimental results are sensitive to the method used to purify astrocytes. For example, Fig. 6b was done with astrocytes purified using the conventional method, while Fig. 6c-e used immunopanning purified cells. The purification conditions are stated in the figure legends if immunopurified astrocytes were used.

All the n values were now specified in figure legends.

2. There are some incorrect statements in the introduction and discussion about tauopathies including Page 3 "intracellular neurofilaments (IFLs)" are described as a pathological hallmark of neurodegenerative diseases that are formed by hyperphosphorylated tau. Neurofibrillary tangles (NFTs) are the hallmark formed by hyperphosphorylated tau in AD. Neurofilaments may be associated with these lesions but are not their primary component. Similarly on Page 14, line 279 "tauopathies" are not found within glial cells, rather tau aggregates or neurofibrillary pathology accumulates in glia in some tauopathies.

RE: We have revised the text on page 3 and page 15 accordingly.

3. The experiments rely on use of tagged recombinant tau. the relevance of heparin-

induced tau filaments has been questioned due to differences with brain-derived tau (Zhang et al., 2019 DOI: 10.7554/eLife.43584). Confidence would be improved if key findings were replicated using filamentous and monomeric tau extracted from postmortem human brain or from transgenic mouse models of tauopathy.

RE: Please note that with the exception of the APEX2-mediated labeling experiments, all other experiments were conducted with untagged Tau monomer or PFFs.

The reviewer's concern on heparin-induced tau filament is important. However, since the integrin receptor was initially identified using monomeric Tau as a bait and because both monomeric Tau and Tau fibrils can activate astrocytes (only to different levels), we believe that our findings cannot be attributed to heparin.

We agree that it is important to confirm our findings with Tau derived from human or mouse tissues, but we feel that this is beyond the scope of the current study.

4. The effect on astrocyte viability and proliferation in response to tau monomer and PFF, shRNA and kinase inhibitors, particularly for experiments with prolonged incubation times (including Fig. 3-6 and presumably 7) should be examined. This is critical for the proper interpretation of the results gained.

RE: Please note that most of the treatment procedures were quite short (1-2h for ice treatment or 6-8h for Tau PFF and kinase inhibitor treatment). We feel that these short treatments should not affect the viability of the cell. Nevertheless, we did the experiments suggested by the reviewer. As you can see from Supplementary Figure 1c, 2c, 3c, these treatments did not induce detectable cell death. We also tested the effect of shRNA knockdown, which took 72 h. The result shown in Supplementary Fig. 3b suggests that knockdown of integrin $\alpha V/\beta 1$ did not affect cell viability either. After prolonged incubation after treatment with Tau PFF for 6 h, we did observe a change in cell morphology (Supplementary Fig. 6a), which is consistent with astrocyte activation, but no significant cell death was observed.

5. The effects of cell incubation on ice for 2/3 hours on cell viability/membrane integrity should be shown (Figs S2, 2). Why were some incubations for 3 hours (Fig. S2A) and others for 2 hours (Fig. S2B, Fig. 2)?

RE: At the beginning, we progressively shortened the incubation time and found that there was no significant impact on Tau binding to the cell surface. In the final pulldown experiment, we chose to incubate astrocytes with Tau on ice for only 30 min (please see the method). We tested the effect of lowering the temperature on astrocyte viability and found no effect on cell viability (Supplementary Fig. 2c).

6. Fig. 1. Since monomeric tau is shown to bind to ITGaV/b1, can an interaction of endogenous tau with endogenous ITGaV/b1 be demonstrated in astrocytes?

RE: This experiment is unlikely to work since extracellular Tau binds to the ectodomain of ITGaV/b1 on the cell surface. Endogenous Tau is cytosolically localized and therefore is compartmentally segregated from the ITGaV/b1 ectodomain. Besides, previous studies suggested that Tau expression is restricted to neuron in the CNS (<https://pubmed.ncbi.nlm.nih.gov/3930508/>)

7. Fig. 4. The NFkappaB findings would be strengthened by analysis of NFkappaB and phospho-NFkappaB protein amounts, in addition to effects of treatment on NFkappaB translocation to the nucleus.

RE: In new Fig. 5e-g, we show by imaging that Tau PFF treatment increases nuclear NFkB while Tau monomer has a weaker effect. **Importantly, the upregulation of inflammation markers by Tau PFF can be reversed by pre-treating cells with a NFkB inhibitor (Fig. 5h,i, Supplementary Fig. 4b).** These results strengthened our conclusion that Tau PFF activates astrocytes via NFkB activation.

8. Fig 6C-D. It would be interesting to see if A2 transcript changes are reciprocal to those found for the A1 transcripts. This would help determine if astrocyte phenotype is recovered in these conditions.

RE: We performed qRT-PCR experiments to analyze the A2 genes. We find that only 3 A2 genes were activated by Tau PFF. Interestingly, the activation of these three genes was similarly reduced when we knocked down ITGaV/b1 or Talin1 (Supplementary Fig. 5). Thus, essentially all gene expression changes induced by Tau PFF could be attributed to ITGaV/b1 activation. The new data also suggest that the activation state of our astrocyte in the presence of Tau PFF is not entirely identical to that reported in Liddelow, S.A. et al. We therefore use the term “A1-like state” or simply call it an “activated state” to describe the astrocyte functional state after Tau PFF treatment.

9. Fig. 7. The viability and numbers/density of treated and replated astrocytes should be assessed. It would be useful to include a cytoskeletal marker in 7B to show the outline of neurons.

RE: We now use the calcein-AM dyes to examine cell viability after re-plating. No significant cell death was observed, but we did notice that Tau PFF-treated cells after replating adopted a different morphology, consistent with their activation state. Because we would like to give readers a more complete view on the level of cell death, we did our imaging experiments using a low magnification. In addition, the calcein-AM dye we used has a broad fluorescence spectrum that can be viewed by other channels including the CFP and blue DPAI channels. This makes it difficult to include a cytoskeletal marker to show the outline of the neurons.

To show the outline of the neurons, we now plated the neurons at a lower density. In this experiment, if we enlarge the area where the imaging chamber is not entirely covered by the cells, we could see some neuronal processes. These pictures are now shown in Supplementary Figure 6.

Minor issues:

1. Page 6. Cite previous references showing that tau primarily binds to the plasma membrane via its N-terminus (e.g.Brandt et al., 1995) and Pooler et al., 2012).

RE: We thank the reviewer for pointing this out. These references are now added.

2. Fig. 1. Define WEC in the legend.

RE: This is now defined in the legend.

3. Fig. 1F legend. Add the concentration of Tau-APEX2 used, and the time of incubation with cells, to the legend.

RE: This is now added.

4. Fig. 2B. Correct typos in figure headings (IGT).

RE: Thank you for noticing the typos. We have fixed the errors.

5. Page 9, line 180. Add references to support the statement that "...fibronectin....downstream genes."

RE: We have added a reference.

6. Page 9, line 183. Correct typo "phosphate-buffered saline" not "phospho-buffered"

RE: This is corrected.

7. Fig 3D: define what is meant by experimental pairs. Is n=2 here?

RE: We have added another experimental repeat here. Experiment pairs were used to show the reproducibility of our experiment. We now provide the p values from unpaired student t-test, n=3. The paired lines are removed.

8. S. Figure 3. Why does recombinant tau yield bands lower than its predicted molecular weight (75kDa) in these IPs?

RE: The Tau variant used here is untagged 2N4R (page 5), which has 441 amino acids. The predicted molecular weight for this protein is 46kD. From published results, this Tau variant usually migrates between 50kD and 75kD (see an example below from <https://www.biorxiv.org/content/10.1101/163394v2.full>), which is consistent with what we see. Our Apex2-tagged Tau does migrate at 75kD because of the addition of the APEX2 tag.

9. Fig. 7. Add info about the age (DIV) or primary neurons to the figure legend and the amount of CM added.

RE: The information is now added. For the CM, we do not have a good way to control the precise amount of the neurotoxic factor in it because its identity is unknown. We tried to start with the same number of astrocytes and follow the same protocol (see the method).

10. Discussion, Pg 14, line 283. Reference 57 does not study propagated tau, only expression of tau in astrocytes. Please correct the wording of this sentence.

RE: This is fixed now on page 15.

Reviewer #2 (Remarks to the Author):

In this manuscript by Wang & Ye, they provide a mechanism by which Tau in both monomeric and pre-formed fibrillar form can drive a reactive state in astrocytes. The authors promote the hypothesis that this reactive state is similar to the recently described neurotoxic/A1 inflammation-induced reactive astrocyte subtype, and further show that secreted factors from these Tau-mediated reactive astrocytes is toxic to neuron cultures. Data to support their model shows involvement of α V β 1 integrin as a receptor for Tau on the surface of astrocytes, and downstream activation pathways including NF- κ B and FAK-associated inflammation pathways. The authors are to be applauded for using cell culture methods that exclude serum in a number of experiments - as this is particularly important for studying immune and glial cell function.

Overall this is a novel pathway that differs from the original inflammation-microglia-astrocyte-neurotoxin axis that was described, and provides some interesting insights into the possibility for multiple mediators of a common reactive state.

RE: We thank the reviewer for his/her appreciation of our work.

I do have a few concerns with this study that would benefit from clarification:

1. production of 'A1'/neurotoxic reactive astrocytes using Tau. This unfortunately was the weakest section of the manuscript. There is no doubt that Tau (either in monomeric or fibrillar form) is driving transcriptomic changes in astrocytes, however whether this is the same as the previously published neurotoxic subtype remains unclear. The original description showed alteration in several cassettes of genes (A1/A2 as described here, but also 'PAN' reactive genes that were common to several reactive subtypes - e.g. Gfap, Lcn2, among others). The smaller subset of genes shown here may, or may not, show a faithful recapitulation of this response (indeed if they do not, but instead promote a different reactive state with an equally toxic function - this would still be very interesting). This could be addressed by analysis of PAN reactive genes from the original publication (REF 30 in this current manuscript), or by unbiased RNASeq to determine other important transcriptomic changes.

RE: We thank the reviewer for this excellent suggestion. We completely agree that the selected A1 genes may not faithfully report the precise activation state of our astrocytes. We now included the pan-reactive genes as well as the A2 specific genes in our qRT-PCR analyses. As you can see from the new Fig. 6a, many of the pan-reactive genes are indeed activated by Tau PFF and to a lesser extent by monomeric Tau. In addition, we also analyzed the effect of ITG α V β 1 and Talin1 knockdown on the induction of pan-reactive and A2 genes and found that essentially all activated genes are diminished by these genetic manipulations (Supplementary Fig. 5). Because we cannot say that our astrocytes activated by Tau PFF are in the exact same state as that reported by Liddel, S.A. et al, we now call it either an "A1-like" state or simply an activated state.

This is also true for the very interesting result (around line 259) that shows the addition of

PF-5622 can decrease the upregulation of many reactive astrocytes genes. It would like to see it explored further - actually looking at all the gene expression changes in Tau-treated astrocytes, and how the PF-562271 changes this.

RE: As mentioned above, we now examined the effect of ITGaV/b1 and Talin1 knockdown on the gene expression changes (Pan reactive, A1 and A2). We feel that this should be sufficient to address the question here, so we did not repeat the PF-5622 experiments with all the genes.

2. neurotoxic capacity. While it is apparent that PFF>monomeric>no Tau astrocyte conditioned media is able to kill neurons in a culture dish, there is no Tau only controls included in these studies. As the authors comment on line 48/49, filamentous Tau itself has been shown to be toxic to neurons - this should be removed from the conditioned media before completing these studies, or compared to a set of Tau only controls.

RE: We are aware of the potential toxicity by Tau, which is why we replated the astrocytes after extensive washing with PBS (see the method on page 26). This procedure involved the use of trypsin to digest any proteins that remain bound to the cell surface. We believe that these washing and replating steps should help us avoid Tau carryover. We did not do the no-cell control because our ITGaV/b1 knockdown clearly rescued this neurotoxic phenotype. We believe that this result excludes the possibility that cell death is caused by Tau carryover (if it were due to Tau carryover, ITGaV/b1 knockdown in astrocytes should not matter).

3. the authors have not considered an internally-regulated function of Tau (by astrocyte phagocytosis) that could mediate some form of reactive response. Can the authors comment on this, and is it possible to study the effects of Tau binding only to aV/B1 integrin by blocking astrocyte phagocytosis?

RE: We now include a statement on page 16 to discuss this possibility, which is clearly important and await further study.

Minor concerns:

1. line 75 - there is also evidence that inflammation-induced astrocyte reactive states have decreased phagocytic capacity

RE: We change "increase" to "alter".

2. line 82 - the original description of A1/neurotoxic astrocytes (REF 30) specifically highlighted that they did not lost neuro-supportive functions, but that they gained a specific neurotoxic capacity that overcame this support.

RE: We have revised our text accordingly.

3. The authors note (paragraph starting at line 190) that traditional serum-containing

cultures of astrocytes often have many contaminating cells, hence their validation of results using immune-panned astrocytes. The purity of these cultures would also be nice to know.

RE: We did check the purity by staining these cells with an antibody against the astrocyte marker GFAP. A representative picture was included in Figure 3c. As far as we could tell, all cells purified by this procedure were positively stained by GFAP antibodies.

4. there are a small number of grammatical/sentence structure errors in the manuscript (e.g. line 64 'on the uptake side')

RE: We have removed this phrase. We also carefully checked the manuscript and tried our best to fix any grammatical mistakes.

5. Figures - please label micrographs with antibodies/fluorescent channels being imaged (e.g. Fig 2f/h) and add missing scale bars (e.g. Fig 3)

RE: We have added fluorescent channel label to Figure 2f/h and scale bar to Fig. 3c.

6. as per convention, gene names should be italicized (in text, and figures)

RE: We have changed the gene names accordingly.

Reviewer #3 (Remarks to the Author):

The authors use a proximity ligation assay to identify a receptor for tau on primary astrocytes in vitro, identifying the integrin receptor complex (ITG α V/ β 1). The authors also find that the receptor signals through Talin and FAK, leading to A1-like astrocyte activation and downstream neuron death. This finding is significant because it identifies a mechanism by which astrocytes internalize misfolded tau protein and could thus contribute to pathology during tauopathy. This has the potential to influence tauopathy/neurodegeneration fields, but also represents a novel astrocyte-neuron communication mechanism since tau release increases with neuron firing. The study appears to use appropriate statistical tests and includes methods that would allow for other researchers to reproduce the work.

Overall the work well done but I do have some concerns:

1. How did you ensure that the recombinant tau was endotoxin-free? As mentioned previously LPS activated astrocytes increase phagocytosis and cytokine/chemokine release etc. This is essential to validate all of the results from the paper. It could be helpful to do an experiment with human or mouse-derived PFFs or seeds isolated, for example, by size exclusion chromatography, to ensure this is not an effect of using E Coli to produce the tau.

RE: There are two steps during the Tau protein purification that ensure no LPS contaminant is present in our protein samples. First, the Tau protein was purified from heat-treated *E. coli* extracts (boiling for 20min) because Tau is heat stable. On the other hand, it has been established that LPS is quite sensitive to heat (Gao et al., J. Leukoc Biol., 2006, [https://pubmed.ncbi.nlm.nih.gov/16720829/#:~:text=Abstract,of%20lipopolysaccharide%20\(LPS\)%20contamination.&text=Heat%20treatment%20by%20boiling%20for,LPS%20TNF%20alpha%20inducing%20activity](https://pubmed.ncbi.nlm.nih.gov/16720829/#:~:text=Abstract,of%20lipopolysaccharide%20(LPS)%20contamination.&text=Heat%20treatment%20by%20boiling%20for,LPS%20TNF%20alpha%20inducing%20activity)). Second, we used two additional chromatography steps to further purify Tau, first by an ion exchange column followed by a gel filtration step. It is unlikely that LPS is still present in our protein sample given its small molecular weight. The fact that Tau PFF is a stronger inflammation inducer than monomeric Tau also suggests that the effect cannot be attributed to LPS (If due to LPS, monomeric Tau should have more contaminant because Tau PFF was further purified by dialysis).

2. The authors suggest that tau-integrin signaling acts via STAT3 and NFkB, but the methods used to assert this are weak. Expression levels of nfkB or STAT3 mRNA (as used here) is not really a readout of NFkB activation. I recommend looking at I κ B α degradation and/or phospho-p65 accumulation and/or nuclear translocation by Western blot. Phospho STAT3 Western blot also would be helpful. Strongest evidence could be through luciferase reporter assay for these TFs with astrocyte PFF treatment.

RE: We thank this reviewer and also reviewer 2 for raising this important issue. We now performed immunostaining using a p65 antibody to check whether NFkB is translocated into the nucleus in cells treated with Tau PFF and monomeric Tau. We detected a modest activation as indicated by an increase in the nuclear/cytoplasmic p65 ratio (Fig. 5e-g). This

phenotype was also observed in Tau monomer-treated cells. The activation of NFkB, albeit modest, is critical for the induction of inflammation and the activation of astrocyte because pre-treatment of cells with a NFkB inhibitor (PDTC) significantly inhibits the induction of inflammation genes by Tau PFF (Fig. 5h, I, Supplementary Fig. 4). Additionally, PDTC treatment also inhibits the induction of A1 genes by Tau PFF (Fig. 6e).

3. The authors suggest that tau-integrin signaling leads to A1 astrocyte activation. It would be helpful to see a positive control of known A1 inducers to compare the effect of tau monomers and PFFs.

RE: As suggested by reviewer 2, we have now conducted a thorough analyses on the expression of both pan reactive, A1, and A2 specific genes in Tau PFF-treated astrocytes as well as in cells with IgtaV/b1 or Talin1 knockdown. Our new data suggest that astrocytes are activated to a state that is not completely identical to the reported A1 state. We now revise the text to term this as either an “A1-like” state or simply an activated state. For this reason, we do not feel the necessity to repeat what has been reported in Liddelov S.A. et al., Nature 2017.

Reviewers' Comments:

Reviewer #1:

Remarks to the Author:

The authors are thanked for their responses which have addressed the main concerns. They have made several important revisions which improve the manuscript. This is an interesting study that will be of great value to the field.

A few final comments/points are:

1. I had assumed that each n represented an independent astrocyte preparation, as is correct, and appreciate the authors clarification about this. Some experiments still have only n=2, which in my opinion is insufficient to consider a finding robust. This applies to Fig. 2h-i, 5g (strange since the experiments from which this data was obtained seem to have been conducted more than twice based on the other data shown?). N is missing for Figure 7.
2. There is evidence from several gene expression studies that astrocytes express tau, although this seems to be particularly true for human as compared to rodent astrocytes. Endogenous, physiological (soluble) species of tau are released, at least from neurons (Pooler et al., 2013; Yamada et al., 2013), and are believed to have intercellular signaling roles (e.g. Avila et al., 2014), hence the question previously raised in Point 6 of my original review. It is interesting that monomeric tau was almost as efficient as fibrillar tau at inducing NFkappaB activation/trafficking to nuclei. This is worthy of further discussion in light of the proposed functional roles of monomeric tau.
3. The authors are thanked for their clarification about recombinant versus brain derived tau. The key concern here is the recombinant nature of the tau used and not the heparin tag. I appreciate that repeating these studies with brain-derived tau is a lot more work and may not be feasible at this time.

Reviewer #2:

Remarks to the Author:

The authors have addressed the concerns originally raised by this reviewer. The quality of the data, and the interpretation of this data is now of a higher standard expected of this journal.

I would however caution the authors about their interpretation of what constitutes an appropriate 'n' value. While individual preparations of astrocytes may have been made using a combination of individual animals, these combined animals now represent a single n value/sample. One cannot use a combined number of animals as a biological replicate when the samples have been combined. I am satisfied that this has not occurred in the representation of the data, but the authors should be careful in using such reasoning in their rebuttal.

The inclusion of additional transcripts to more comprehensively show the activation state of astrocytes following treatment with Tau species is also a nice addition to the manuscript.

For Tau carryover concerns - while the authors reasoning is sound, as a reader/reviewer I would be more comfortable with this conclusion if this was explicitly shown. Does trypsin/PBS washing remove Tau completely - a western blot/immuno stain would clearly show this and there would be no concern with the following experiments or interpretation of these results. This still seems like an important control to show the reader.

Additions to Figure 3 about culture purity are appreciated. Thank you.

I would also like to state I was particularly thankful for the comments to Reviewer #3 about endotoxin removal from Tau species used in these experiments. I had missed this in the original review of the manuscript, and the methodology the authors have employed to ensure no carryover

of endotoxin from bacterial production of the recombinant protein is fantastic.

Reviewer #3:

Remarks to the Author:

The authors use a proximity ligation assay to identify a receptor for tau on primary astrocytes in vitro, identifying the integrin receptor complex (ITG α V/ β 1). The authors also find that the receptor signals through Talin and FAK, leading to A1-like astrocyte activation and downstream neuron death. The initial manuscript did not have strong data indicating the specific pathway that Talin and FAK might use, but suggested NF κ B and/or STAT3 pathways with weak supporting data. The revised manuscript has greatly improved the supporting data suggesting an NF κ B-dependent mechanism, using confocal imaging of translocation and an NF κ B inhibitor. The authors also improved their analysis of astrocyte phenotype in the revised version, which strengthens the paper. The findings are significant because the authors identify a mechanism by which astrocytes internalize misfolded tau protein, adopt and activated neurotoxic state and could thus contribute to pathology during tauopathy. The strengthened manuscript implicating NF κ B improves the translational potential of this finding. This has the potential to influence tauopathy/neurodegeneration fields, but also represents a novel astrocyte-neuron communication mechanism since tau release increases with neuron firing. The study appears to use appropriate statistical tests and includes methods that would allow for other researchers to reproduce the work. While not within the scope of the current paper, I hope that the authors consider extending their findings to tau derived from mouse or human sources in future studies.

Overall, the revisions strengthen the paper significantly and I recommend that the paper be accepted pending confirmation from other reviewers that their concerns were addressed sufficiently.

REVIEWER COMMENTS

Reviewer #1

The authors are thanked for their responses which have addressed the main concerns. They have made several important revisions which improve the manuscript. This is an interesting study that will be of great value to the field.

RE: We thank the reviewer for the positive feedback.

A few final comments/points are:

1. I had assumed that each n represented an independent astrocyte preparation, as is correct, and appreciate the authors clarification about this. Some experiments still have only n=2, which in my opinion is insufficient to consider a finding robust. This applies to Fig. 2h-i, 5g (strange since the experiments from which this data was obtained seem to have been conducted more than twice based on the other data shown?). N is missing for Figure 7.

RE: We have repeated the experiment in Fig. 2h, i and Figure 5g. These experiments are now repeated 3 times independently. We also repeated the experiment in Figure 7 three independent times. We now add the information to the figure legends.

2. There is evidence from several gene expression studies that astrocytes express tau, although this seems to be particularly true for human as compared to rodent astrocytes. Endogenous, physiological (soluble) species of tau are released, at least from neurons (Pooler et al., 2013; Yamada et al., 2013), and are believed to have intercellular signaling roles (e.g. Avila et al., 2014), hence the question previously raised in Point 6 of my original review. It is interesting that monomeric tau was almost as efficient as fibrillar tau at inducing NFkappaB activation/trafficking to nuclei. This is worthy of further discussion in light of the proposed functional roles of monomeric tau.

RE: I now see that the reviewer suggested to test whether endogenous Tau released by neurons can interact with integrin on the surface of astrocytes. We agree that this is an important question. However, this experiment is technically challenging because the level of endogenous Tau secretion is very low (usually less than 1% of the total Tau). For example, in the study by Pooler et al., 2013, only a weak Tau signal was detected by immunoblotting in the conditioned medium after 50-fold concentration.

As for the role of Tau in NFkappaB activation, the reviewer raises an important point, which is now discussed in the paper. It is noteworthy that there is a small but statistically significant difference in the level of NFkB translocation between cells treated with monomeric Tau and Tau PFF (Fig. 5g). Additionally, our mass spectrometry data suggests that ITGAV/b1 is NOT THE ONLY receptor that Tau binds on the cell surface. Thus, the integrin-NFkappaB signaling axis should not be the only signaling output from Tau binding. We propose that it is the combination of the NFkB activation levels and other additional signaling inputs that collectively shape the outcome of astrocyte activation. This is now discussed on page 16.

3. The authors are thanked for their clarification about recombinant versus brain derived tau. The key concern here is the recombinant nature of the tau used and not the heparin tag. I appreciate that repeating these studies with brain-derived tau is a lot more work and may not be feasible at this time.

RE: We agree with the reviewer that repeating the study with brain-derived Tau is beyond the scope of the current study.

Reviewer #2 (Remarks to the Author):

The authors have addressed the concerns originally raised by this reviewer. The quality of the data, and the interpretation of this data is now of a higher standard expected of this journal.

RE: Thank you for the positive evaluation.

I would however caution the authors about their interpretation of what constitutes an appropriate 'n' value. While individual preparations of astrocytes may have been made using a combination of individual animals, these combined animals now represent a single n value/sample. One cannot use a combined number of animals as a biological replicate when the samples have been combined. I am satisfied that this has not occurred in the representation of the data, but the authors should be careful in using such reasoning in their rebuttal.

RE: We totally agree with this reviewer. The only reason for mentioning animal number in our previous rebuttal letter was because we were not sure why reviewer 1 thought that "The sample size of most experiments is very small (n=3)....". For cell biology papers in NC and other journals, 3 independent experiments is the norm, while some papers even had most experiments done only twice (n=2) (e.g. <https://www.nature.com/articles/s41467-020-15000-w>). We thought that the reviewer 1 might mistakenly compare our study to animal studies, which often requires analyzing phenotypes in at least 8 animals (e.g. n=8) due to individual variations. We did consider the experimental variations from animal to animal, which is why we pooled a few animals in each astrocyte preparation, but we only counted that as one independent experiment. We tried to clarify this point in our letter, but we did not mean to use this to argue for a larger n number.

The inclusion of additional transcripts to more comprehensively show the activation state of astrocytes following treatment with Tau species is also a nice addition to the manuscript.

RE: Thank you for the comment.

For Tau carryover concerns - while the authors reasoning is sound, as a reader/reviewer I would be more comfortable with this conclusion if this was explicitly shown. Does trypsin/PBS washing remove Tau completely - a western blot/immuno stain would clearly show this and

there would be no concern with the following experiments or interpretation of these results. This still seems like an important control to show the reader.

RE: As suggested, we have repeated the astrocyte treatment experiment and performed Western blot. As you can see from the data below, we could only detect Tau in the conditioned medium right after Tau-treatment. After we transferred astrocytes to a new plate, the second conditioned medium (CM2) contains no Tau.

Astrocytes were treated with PBS, monomeric Tau (200nM) or Tau PFF (200nM) for 6h. Conditioned medium (CM1) was harvested. Cells were then washed, trypsinized, and transferred to a new plate. After cell attachment, conditioned medium (CM2) was harvested. The CM1 and CM2 were analyzed by immunoblotting together with purified Tau monomer and PFF as a control.

Additions to Figure 3 about culture purity are appreciated. Thank you.

I would also like to state I was particularly thankful for the comments to Reviewer #3 about endotoxin removal from Tau species used in these experiments. I had missed this in the original review of the manuscript, and the methodology the authors have employed to ensure no carryover of endotoxin from bacterial production of the recombinant protein is fantastic.

RE: Thank you for the positive comments.

Reviewer #3 (Remarks to the Author):

The authors use a proximity ligation assay to identify a receptor for tau on primary astrocytes in vitro, identifying the integrin receptor complex (ITG α V/ β 1). The authors also find that the receptor signals through Talin and FAK, leading to A1-like astrocyte activation and downstream neuron death. The initial manuscript did not have strong data indicating the specific pathway that Talin and FAK might use, but suggested NF κ B and/or STAT3 pathways with weak supporting data. The revised manuscript has greatly improved the supporting data

suggesting an NFkB-dependent mechanism, using confocal imaging of translocation and an NFkB inhibitor. The authors also improved their analysis of astrocyte phenotype in the revised version, which strengthens the paper. The findings are significant because the authors identify a mechanism by which astrocytes internalize misfolded tau protein, adopt and activated neurotoxic state and could thus contribute to pathology during tauopathy. The strengthened manuscript implicating NFkB improves the translational potential of this finding. This has the potential to influence tauopathy/neurodegeneration fields, but also represents a novel astrocyte-neuron communication mechanism since tau release increases with neuron firing. The study appears to use appropriate statistical tests and includes methods that would allow for other researchers to reproduce the work. While not within the scope of the current paper, I hope that the authors consider extending their findings to tau derived from mouse or human sources in future studies.

Overall, the revisions strengthen the paper significantly and I recommend that the paper be accepted pending confirmation from other reviewers that their concerns were addressed sufficiently.

RE: Thank you for the positive feedback. We will for sure consider this suggestion in our future study.

Reviewers' Comments:

Reviewer #1:

Remarks to the Author:

The authors have addressed all of my concerns, and I support publication of this work.

Point-by-point response to reviewers:

REVIEWERS' COMMENTS

Reviewer #1 (Remarks to the Author):

The authors have addressed all of my concerns, and I support publication of this work.

RE: We thank the reviewer for his support of the publication of our work.